# IMPROVED APPROXIMATION ALGORITHMS FOR $k$-SUBMODULAR MAXIMIZATION VIA MULTILINEAR EXTENSION

**Huanjian Zhou**[1,2] **Lingxiao Huang**[3] **Baoxiang Wang**[4*]

[1]The University of Tokyo [2]RIKEN AIP

[3]State Key Laboratory of Novel Software Technology, New Cornerstone Science Laboratory,
 Nanjing University

[4]The Chinese University of Hong Kong Shenzhen

## ABSTRACT

We investigate a generalized form of submodular maximization, referred to as $k$-submodular maximization, with applications across the domains of social networks and machine learning. In this work, we propose the multilinear extension of $k$-submodular functions and unified Frank-Wolfe-type frameworks based on that. This continuous framework accommodates 1) monotone or non-monotone functions, and 2) various constraint types including matroid constraints, knapsack constraints, and their combinations. Notably, we attain an asymptotically optimal $1/2$-approximation for monotone $k$-submodular maximization problems with knapsack constraints, surpassing previous $1/3$-approximation results (Ha et al., 2024), and a factor-$1/3$ approximation for non-monotone $k$-submodular maximization problems with knapsack constraints and matroid constraints which outperforms previous $0.245$-approximation results (Yu et al., 2023). The foundation for our analysis stems from new insights into specific linear and monotone properties pertaining to the multilinear extension.

## 1 INTRODUCTION

Consider the following problems in machine learning and operations research: (i) identifying influential individuals in a social network with $k$ topics to maximize the number of individuals influenced by at least one topic (Qian et al., 2017; Zhang et al., 2019), (ii) partitioning a set of features into $k + 1$ subsets such that one feature can be used in at most one regression target (or none of them) for $k$ regression targets on these features (Singh et al., 2012b; Zhou et al., 2019), and (iii) selecting a small set of sensors from $k$ types of sensors in an area to maximize the information obtained from the sensors (Ohsaka & Yoshida, 2015). These problems are often constrained, such as selecting sensors with different costs within a finite budget. More examples can be found in Appendix A.1.

Solving these problems involves maximizing a $k$-*submodular* set function $f : \{0, \ldots, k\}^n \to \mathbb{R}_{\geq 0}$ subject to some constraints. Intuitively, the $k$-submodularity property captures the notion of diminishing returns. For instance, consider identifying influential individuals in a social network. For a fixed topic, the newly selected influential individuals will contribute less to the overall user coverage if many influential individuals have already been selected, and more if only a few have been selected. Similarly, adding additional features in regression problems and placing additional sensors in an area also share such diminishing returns property.

Formally, for an integer $k \geq 1$ and a finite nonempty set $[n]$, a non-negative function $f : \{0, \ldots, k\}^n \to \mathbb{R}_{\geq 0}$ is called $k$-submodular if for all $\mathbf{s}$ and $\mathbf{t}$ in $\{0, \ldots, k\}^n$, we have

$$f(\mathbf{s}) + f(\mathbf{t}) \geq f(\min_0(\mathbf{s}, \mathbf{t})) + f(\max_0(\mathbf{s}, \mathbf{t})), \tag{1}$$

---

[*]Corresponding to: Huanjian Zhou, Lingxiao Huang, Baoxiang Wang

where for every $i \in [n]$,

$$\min_0(\mathbf{s}, \mathbf{t})_i = \begin{cases} 0, & \mathbf{s}_i \mathbf{t}_i \neq 0 \text{ and } \mathbf{s}_i \neq \mathbf{t}_i, \\ \min(\mathbf{s}_i, \mathbf{t}_i), & \text{otherwise,} \end{cases} \quad \text{and}$$

$$\max_0(\mathbf{s}, \mathbf{t})_i = \begin{cases} 0, & \mathbf{s}_i \mathbf{t}_i \neq 0 \text{ and } \mathbf{s}_i \neq \mathbf{t}_i, \\ \max(\mathbf{s}_i, \mathbf{t}_i), & \text{otherwise.} \end{cases}$$

We provide an illustrative example of sensor placement for $k$-submodular: Suppose there are $k = 3$ types of sensors and $n = 4$ different locations to deploy them. The vector $\mathbf{s} = (1, 2, 0, 3)$ means placing type-1, type-2, no sensor, type-3 sensor, in locations 1, 2, 3, 4, respectively. The vector $\mathbf{t} = (0, 2, 1, 3)$ means placing no sensor, type-2, type-1, type-3 sensor, in locations 1, 2, 3, 4, respectively. Then $\min_0(\mathbf{s}, \mathbf{t}) = (0, 2, 0, 3)$ and $\max_0(\mathbf{s}, \mathbf{t}) = (1, 2, 1, 3)$. Intuitively, adding more sensors to a single task does not yield proportional benefits compared to evenly distributing them across tasks. Such diminishing return property is described by Eq. (1) that $f(1, 2, 0, 3) - f(0, 2, 0, 3) \geq f(1, 2, 1, 3) - f(0, 2, 1, 3)$. In fact, the diminishing return applies to every coordinate of the problem by the inequality. Our definition of $k$-submodular functions (Eq. (1)), which is also employed in Ward & Zivný (2016), is equivalent to an alternative definition used in Iwata et al. (2016); Sakaue (2017). In this paper, we use the definition of Eq. (1) since it is more convenient to define and study the properties of multilinear extension. For completeness, we give the explicit form of the equivalent definition and show their equivalence in Appendix A.2.

A special case of the $k$-submodular maximization problem is the submodular maximization problem with $k = 1$. The techniques for submodular maximization problems can generally be classified into two main lines. The first line is combinatorial and is mostly based on greedy rules and local search. This approach has been applied to both monotone and non-monotone submodular objective functions under various constraints (Buchbinder et al., 2015; Feige et al., 2011; Filmus & Ward, 2014; Lee et al., 2010a;b; Nemhauser et al., 1978). In some cases, optimal algorithms have been obtained using this line of approaches (Buchbinder et al., 2015; Sviridenko, 2004). The second line is a two-staged framework based on the multilinear extension. This line of methods involves identifying a fractional solution for the relaxation of the problem and then rounding the fractional solution to obtain an integral one while incurring a bounded loss in the objective. This line of approaches achieves better approximation ratios in most cases (Buchbinder & Feldman, 2019; Călinescu et al., 2011; Chekuri et al., 2010a; 2014; Feldman et al., 2011a; Kulik et al., 2013).

Previous works in constrained $k$-submodular function maximization were based on combinatorial techniques, such as the greedy algorithm. However, compared with tight approximations of submodular maximization with various constraints, previous combinatorial approaches have not been able to achieve asymptotically optimal approximation results in most cases. In fact, the only tight results available are on the basic case of a single matroid constraint. For example, even for the important case of monotone $k$-submodular maximization with single knapsack constraint, the current best combinatorial method only obtains $1/3$-approximation (Ha et al., 2024), leaving a large gap with the best known lower bound of $\frac{k+1}{2k}$ (Iwata et al., 2016). Also, existing combinatorial methods do not provide the flexibility to combine constraints of different types, especially $O(1)$ knapsack constraints.

## 1.1 OUR CONTRIBUTIONS

We propose a generalization of multilinear extension for the case of $k$-submodular functions (Definition 2.4), which allows us to use continuous optimization methods for this class of problems. Using this new concept of multilinear extension, we propose a unified framework, that is capable of handling different types of constraints, as well as monotone and non-monotone cases (Problem 2.3). We study two classic types of constraints: matroid constraints and knapsack constraints.

We first present the results when $f$ is non-negative and *monotone*, i.e., if $f(\mathbf{s}) \leq f(\mathbf{t})$ holds for every pair of integral vectors $\mathbf{s}, \mathbf{t} \in \{0, \dots, k\}^n$ satisfying that every non-zero coordinate of $\mathbf{s}$ has the same value as that of $\mathbf{t}$, i.e., 1) $\text{supp}(\mathbf{s}) \subseteq \text{supp}(\mathbf{t})$, where $\text{supp}(\mathbf{s}) := \{e \in [n] : \mathbf{s}_e \neq 0\}$ represents the support set, and 2) $\mathbf{s}_e = \mathbf{t}_e$ for all $e \in \text{supp}(\mathbf{s})$.

**Theorem 1.1** (**Informal, see Theorem 3.1**). *For monotone constrained $k$-submodular maximization, there exists a (randomized) polynomial-time algorithm that returns 1) $1/2 - \varepsilon$ approximation for $O(1)$ knapsacks; 2) $1/2 - \varepsilon$ approximation for a single matroid; 3) $0.3/b - \varepsilon$ approximation for the intersection of $O(1)$ knapsacks and $b$ matroids.*

| Problem type of $k$-submod. max. | | Prior results | Our results |
|---|---|---|---|
| $d$ knapsacks $(d = O(1))$ | Monotone | $\frac{1}{3}$ (Ha et al., 2024) $(d = 1)$ $\frac{1}{2+2d} - \varepsilon$ (Gong et al., 2024) | $\frac{1}{2} - \varepsilon$♠ |
| | Non-monotone $(k \geq 2)$ | $\frac{1-e^{-4}}{4}$ (Yu et al., 2023) $(d = 1)$ $\frac{1}{3+2d} - \varepsilon$ (Gong et al., 2024) | $\frac{1}{3} - \varepsilon$ |
| single matroid | Monotone | $\frac{1}{2}$ (Sakaue, 2017)♠ | $\frac{1}{2} - \varepsilon$♠ |
| | Non-monotone $(k \geq 2)$ | $\frac{1-e^{-4}}{4}$ (Yu et al., 2023) | $\frac{1}{3} - \varepsilon$ |
| $b$ matroids + $d$ knapsacks $(d = O(1))$ | Monotone | $\frac{1-e^{-(b+2)}}{b+2}$ (Yu et al., 2023) $(d = 1)$ | $\frac{0.3}{b} - \varepsilon$ |
| | Non-monotone $(k \geq 2)$ | $\frac{1-e^{-(b+3)}}{b+3}$ (Yu et al., 2023) $(d = 1)$ | $\frac{0.2}{b} - \varepsilon$ |

Table 1: Comparison with previous work for constrained $k$-submodular maximization. $\epsilon > 0$ can be a constant arbitrarily close to 0. Symbol ♠ represents that the results are asymptotically tight.

For a single knapsack constraint, we achieve an approximation ratio of $1/2 - \varepsilon$, which aligns with the established lower bound of $\frac{k+1}{2k}$ (Iwata et al., 2016) and is thus asymptotically tight. This outcome represents an improvement over the prior $1/3$-approximation for a single knapsack (Ha et al., 2024). Additionally, we successfully extend this $1/2 - \varepsilon$ approximation ratio to scenarios with $O(1)$ knapsack constraints, maintaining asymptotic tightness. Contrasting with a recent result by (Gong et al., 2024), which provides an approximation ratio of $\frac{1}{2+2d} - \varepsilon$ where $d$ represents the number of knapsack constraints, our result eliminates the dependency factor of $2d$, significantly enhancing the scalability as $d$ increases. In the context of a single matroid constraint, we also secure an asymptotically optimal approximation ratio of $1/2 - \varepsilon$, corroborating the findings of previous studies (Sakaue, 2017).

Furthermore, our algorithm adeptly handles intersections of $O(1)$ knapsacks and $b$ matroids, achieving an approximation ratio of $0.3/b - \varepsilon$. This extends beyond the capabilities of Yu et al. (2023), which addresses only the intersection of a single knapsack and $b$ matroids. We remark that for a single knapsack case, their approximation ratio $\frac{1-e^{-(b+2)}}{b+2}$ is better than our ratio $0.3/b - \varepsilon$ for any $b$. Notably, the factor $1/b$ is justified by a lower bound of $O(\log b/b)$ (Appendix G). A comprehensive summary of these results is available in Table 1.

We then present results when $f$ is non-negative and non-monotone.

**Theorem 1.2** (**Informal, see Theorem F.1**). *For non-monotone constrained $k$-submodular maximization where $k \geq 2$, there exists a (randomized) polynomial-time algorithm that returns 1) $1/3 - \varepsilon$ approximation for $O(1)$ knapsacks; 2) $1/3 - \varepsilon$ approximation for a single matroid 2) $0.2/b - \varepsilon$ approximation for the intersection of $O(1)$ knapsacks and $b$ matroids.*

The theorem presents enhanced approximation ratios for non-monotone objectives under knapsack and matroid constraints. Specifically, for a single knapsack constraint or a single matroid constraint, we have improved the approximation ratio from $(1 - e^{-4})/4$ (approximately 0.25) as established by (Ha et al., 2024), to $1/3 - \varepsilon$ (approximately 0.33). Additionally, for scenarios involving $d = O(1)$ knapsack constraints, we maintain this improved ratio of $1/3 - \varepsilon$, surpassing the recent result of $1/(3 + 2d)$ found in (Gong et al., 2024). Our results eliminate the dependency factor of $2d$, thereby significantly enhancing scalability as $d$ increases.

Furthermore, we extend the improvements beyond the framework of the intersection of a single knapsack and $b$ matroids, as in (Yu et al., 2023), to include intersections of $O(1)$ knapsacks and $b$ matroids, with only minimal reduction in the approximation ratio. See also Table 1 for a summary.

## 1.2 TECHNICAL OVERVIEW

We adopt the idea of two-stage continuous methods of submodular maximization for $k$-submodular problems. Our techniques, however, depart crucially from the works of submodular maximization in the design of multilinear extension for $k$-submodular functions (Definition 2.4) and the utilization of its new properties (Lemma C.1). With the multilinear extension for $k$-submodular functions, we further introduce Frank-Wolfe-type methods (Algorithms 1 and 3) and a novel rounding scheme (Lemma 3.2 and F.3). Below we summarize these notion contributions and technical novelties.

**Recap of multilinear extension for submodular functions**   For a submodular function $f$ : $\{0,1\}^n \rightarrow \mathbb{R}_{\geq 0}$, its multilinear extension $F : [0,1]^n \rightarrow \mathbb{R}_{\geq 0}$ provides a useful relaxation of $f$ to the continuous space $[0,1]^n$. Intuitively, for every item $\bar{i} \in [n]$, a fractional value $\mathbf{x}_i$ in the relaxed continuous space $[0,1]$ represents that the item is selected with probability $\mathbf{x}_i$. Hence, given a fractional point $\mathbf{x} \in [0,1]^n$, $F(\mathbf{x})$ is defined as the expected value of $f(S)$ where each element $i$ is included in the random set $S$ with probability $\mathbf{x}_i$. Such a formulation based on the expectation maintains coordinate linearity and specific submodular properties in a continuous domain. The advantageous properties of multilinear extensions have been leveraged in numerous previous studies, such as Buchbinder & Feldman (2019); Ene & Nguyen (2016); Călinescu et al. (2011); Chekuri et al. (2014), to develop continuous methodologies for constrained submodular maximization. This approach typically unfolds in two distinct phases.

In the continuous optimization stage, the process begins with an empty initial solution $\mathbf{x}(0) = 0$ and progressively updates this solution within the interval $[0,1]$. Through continuous methods, a fractional solution $\mathbf{x}(1) \in [0,1]^n$ is derived, which approximates the maximization of the extension $F$ subject to certain combinatorial constraints. For example, the well-established Frank-Wolfe algorithm (Călinescu et al., 2011) iteratively updates the current solution $\mathbf{x}$ by moving it in the direction of a constrained vector $\mathbf{v}$ that maximizes the local gain, quantified by $\langle \nabla F(\mathbf{x}), \mathbf{v} \rangle$, where $\nabla F(\mathbf{x})$ denotes the gradient vector of $F$ at $\mathbf{x}$. Each iteration aims to enhance the value of $F(\mathbf{x})$ by an amount proportional to $F(\mathbf{o}^\star) - F(\mathbf{x})$, with $\mathbf{o}^\star \in [0,1]^n$ being the vector that optimizes the multilinear extension $F$ under the given constraints. Following the continuous optimization, the rounding stage involves converting the fractional solution $\mathbf{x}(1)$ into a feasible integral solution. In this stage, each element $i$ vies for inclusion based on its coordinate value $\mathbf{x}(1)_i$, thereby rounding $\mathbf{x}(1)$ to an integral form that adheres to the problem's constraints. This two-stage framework not only facilitates the effective handling of combinatorial constraints but also optimizes the solution with respect to the submodular function's multilinear extension.

**Notion contribution: multilinear extension of $k$-submodular functions.**   We introduce the concept of the $k$-multilinear extension for $k$-submodular functions (Definition 2.4), which serves as a natural extension to the multilinear extension for submodular functions. For an item $i \in [n]$, our goal is to represent a probability distribution on the set $\{0, 1, \ldots, k\}$ using a fraction point $\mathbf{x}_i$. To achieve this, we specify the domain of each $\mathbf{x}_i$ as $\Delta_k = \{\mathbf{y} \in [0,1]^k : \sum_{j=1}^{k} \mathbf{y}_j \leq 1\}$, where each coordinate $\mathbf{x}_{i,j}$ indicates the probability of item $i$ being assigned to state $j$, and $1 - \sum_{j \in [k]} \mathbf{x}_{i,j}$ represents the residual probability that item $i$ being assigned to state 0. Subsequently, a $k$-multilinear extension $F$ is defined over the domain $\Delta_k^n$, where $F(\mathbf{x})$ (for $\mathbf{x} \in \Delta_k^n$) computes the expected value of $f(\mathbf{s})$, with each $\mathbf{s}_i = j$ occurring with probability $\mathbf{x}_{i,j}$.

The $k$-multilinear extension exhibits several advantageous properties that facilitate the development of continuous optimization methods. These include multilinearity, element-wise non-positive Hessian, pairwise monotonicity, approximate linearity, and the preservation of monotonicity as delineated in Lemma C.1. The richness of these attributes makes the $k$-multilinear extension a compelling subject of study for $k$-submodular functions and could potentially offer valuable insights independently.

**Challenges of using $k$-multilinear extension.**   Similar to the submodular case, we aim to use the $k$-multilinear extension to design two-staged continuous methods. The extension of the rounding stage is rather straightforward, for both monotone and non-monotone cases (Lemmas 3.2 and F.3). The technical challenges come from the continuous optimization stage, which we summarize below.

- *Closure:* The domain of submodular extension is $[0,1]^n$, which benefits the closure of the coordinate-wise maximum operation, i.e., $\mathbf{x} \vee \mathbf{y} \in [0,1]^n$ for all $\mathbf{x}, \mathbf{y} \in [0,1]^n$. This property is crucial in the analysis of the approximation ratio of the derived fractional solution $\mathbf{x}(1)$. However, the domain of $k$-submodular extension is the corner of the cube $\Delta_k^n$ rather than $[0,1]^{nk}$. Consequently, the closure property no longer holds.

- *Approximate linearity:* Another advantageous property of submodular extensions is their approximate linearity, whereby the function $F$ closely approximates a linear function along any given direction (Bian et al., 2017). This property significantly influences the value change at each step in continuous optimization methods. However, it remains uncertain whether this property extends to $k$-submodular extensions, given the complex structure introduced by the $k$ coordinates.

**Technical novelty.** Our technical contributions lie in tackling these challenges by introducing auxiliary points for analysis and uncovering novel properties of $k$-multilinear extension. We outline these approaches below.

To address the initial challenge about closure, we shift our focus to the operation of linear combination, which constructs $a\mathbf{x} + (1 - a)\mathbf{y}$ (for $a \in [0, 1]$) for $\mathbf{x}, \mathbf{y} \in \Delta_k^n$, rather than using the coordinate-wise maximum for submodular. This operation benefits from closure within $\Delta_k^n$, thereby facilitating the construction of auxiliary points for analytical purposes. More specifically, we consider $\mathbf{o}^\star$ as the optimal fractional solution for the $k$-submodular extension $F$. At each time step $t \in [0, 1]$, we define $\mathbf{x}(t) \in t \cdot \Delta_k^n$ as the current point.[1] We generate a pseudo-convex-combination point $\mathbf{o}(t) = \mathbf{x}(t) + (1 - t)\mathbf{o}^\star$, an auxiliary point that is assuredly within $\Delta_k^n$ due to the set's closure properties. This method of using pseudo-convex-combination points aligns with approaches previously explored in literature, such as Iwata et al. (2016), Ohsaka & Yoshida (2015), and Sakaue (2017). These studies have demonstrated the utility of such points in facilitating detailed and effective analysis.

Then we investigate the relation between $F(\mathbf{x}(t))$ and $F(\mathbf{o}(t))$, whose key is to address the afore-mentioned second challenge about approximate linearity. When $f$ is monotone, we demonstrate that $F(\mathbf{x}(t + \delta)) - F(\mathbf{x}(t)) \gtrsim F(\mathbf{o}(t)) - F(\mathbf{o}(t + \delta))$, which directly leads to a conclusion that $F(\mathbf{x}(1)) \gtrsim \frac{1}{2}F(\mathbf{o}^\star)$ (Lemma 3.3). Theorem 1.1 is a direct corollary of this conclusion and the rounding guarantee (Lemma 3.2). We establish this result based on extending the approximate linearity property for submodular to $k$-submodular functions (Lemma C.1). This property captures certain Lipschitzness of $k$-multilinear extension $F$ and allows us to estimate the increment $F(\mathbf{x}(t+\delta)) - F(\mathbf{x}(t))$ for sufficiently small values of $\delta$.

When $f$ is non-monotone, we utilize a new property of $k$-multilinear extension $F$, called *pairwise monotonicity* (Lemma C.1), which help reduce the problem to the monotone case. Utilizing pairwise monotonicity, we are able to obtain an approximation $F(\mathbf{x}(1)) \gtrsim \frac{1}{3}F(\mathbf{o}^\star)$ (Lemma F.2). Similarly, Theorem 1.2 is a direct corollary of this approximation and the rounding guarantee (Lemma 3.2).

**Comparison with existing combinatorial approaches.** We demonstrate that our approach using continuous optimization methods yields improved approximations for knapsack constraints compared to prior combinatorial methods such as those presented in (Ha et al., 2024; Yu et al., 2023). We offer intuitive explanations for this improvement and observe that a similar conclusion holds for submodular maximization with $O(1)$ knapsack constraints: to the best of our knowledge, no combinatorial method achieves an optimal approximation, whereas an optimal approximation algorithm via multilinear extension has been presented by (Chekuri et al., 2014). Our findings may suggest that the flexibility of continuous methods in selecting stepsizes and update directions provides an advantage over combinatorial approaches for handling knapsack constraints.

### 1.3 Other related works

Submodular maximization, a special case of $k$-submodular maximization, has a rich line of research with numerous results. In the monotone case, tight $(1 - 1/e)$-approximations have been proposed for various constraints, such as single matroid constraint and $O(1)$ knapsacks constraint (Călinescu et al., 2011; Chekuri et al., 2014; Kulik et al., 2009; Nemhauser et al., 1978). Furthermore, additional results have been developed for more complicated constraints, including the intersection of matroids and exchange systems (Feldman et al., 2011b; Lee et al., 2010b). In the non-monotone case, the best-known approximation ratio for the single matroid or $O(1)$ knapsack constraint is $0.401$ (Buchbinder & Feldman, 2024) while the hardness of $0.478$ holds for single matroid (Gharan & Vondrák, 2011).

**Concurrent work** Recent developments in $k$-submodular maximization research have introduced new algorithms with varying approximation ratios. For single matroid constraints, the threshold-decreasing algorithm in Niu et al. (2023) achieves a $1/2$-approximation ratio for monotone objectives and a $1/3$-approximation ratio for non-monotone cases. For single knapsack constraints, an alternative greedy algorithm with $0.432$- and $0.317$-approximation ratios for monotone and non-monotone objectives, respectively, is presented in (Xiao et al., 2023). In comparison, our algorithms outperform these approaches by achieving better approximation ratios or allowing more general types of constraints. For instance, for the non-monotone case with a single constraint, our algorithm

---

[1] $t \cdot \Delta_k^n$ is defined as $\{\mathbf{y} \in \Delta_k^n : \frac{1}{t}\mathbf{y} \in \Delta_k^n\}$.

achieves an approximation ratio $1/3 - \varepsilon$ instead of $0.317$ in (Xiao et al., 2023). Our algorithms also offer greater flexibility across various constraints and achieve a tight $1/2$ approximation ratio for monotone objectives and a $1/3$ approximation ratio for non-monotone objectives with $O(1)$ knapsack constraints and single matroid constraints.

## 2    PROBLEM FORMULATION AND $k$-MULTILINEAR EXTENSION

In this section, we first define the constrained $k$-submodular maximization problem and then present the notion of $k$-multilinear extension. Let $[n]$ be the ground set. Let $f : \{0, 1, \ldots, k\}^n \to \mathbb{R}_{\geq 0}$ be a non-negative $k$-submodular function. Throughout this paper, we assume there exists a value oracle $\mathcal{O}_f$ that answers $f(\mathbf{s})$ for any query $\mathbf{s} \in \{0, \ldots, k\}^n$.

**$k$-submodular maximization with matroid and knapsack constraints.**    As outlined in Section 1, our study encompasses two classic types of constraints: matroid and knapsack constraints. In the context of submodular functions, these constraints are typically defined over the domain $2^n$. Specifically, a constraint for submodular functions is represented by a down-close collection $\mathcal{I} \subseteq 2^n$, where "down-close" means that for any $S \in \mathcal{I}$ and $T \subseteq S$, $T$ also belongs to $\mathcal{I}$.

However, it is important to note that the domain for a $k$-submodular function is not $2^n$ but $\{0, \ldots, k\}^n$. Consequently, in a $k$-submodular setting, it becomes necessary to adapt any given constraint $\mathcal{I}$ to the domain $\{0, \ldots, k\}^n$. This adaptation involves defining the constraint based on the support set $\text{supp}(\mathbf{s})$ of a vector $\mathbf{s} \in \{0, \ldots, k\}^n$, where $\text{supp}(\mathbf{s})$ is defined as the collection of indices corresponding to non-zero coordinates of $\mathbf{s}$. Therefore, a vector $\mathbf{s} \in \{0, \ldots, k\}^n$ is deemed feasible with respect to the constraint $\mathcal{I}$ if and only if its support set, $\text{supp}(\mathbf{s})$, is a member of $\mathcal{I}$. This redefinition ensures that the extended constraints are appropriately applied within the $k$-submodular context.

Using this idea of domain adaption, we provide the following notions of matroid constraints and knapsack constraints for $k$-submodular maximization.

**Definition 2.1** (**Matroid constraint for $k$-submodular maximization**). *Denote a matroid by a pair $\mathcal{M} = ([n], \mathcal{I}_\mathcal{M})$ where $\mathcal{I}_\mathcal{M} \subseteq 2^n$, such that 1) $\forall B \in \mathcal{I}_\mathcal{M}, A \subset B \Rightarrow A \in \mathcal{I}_\mathcal{M}$; 2) $\forall A, B \in \mathcal{I}_\mathcal{M}, |A| < |B| \Rightarrow \exists x \in B \setminus A \text{ s.t. } A \cup \{x\} \in \mathcal{I}_\mathcal{M}$. A matroid constraint for $k$-submodular is defined to be the collection of all vectors $\mathbf{s} \in \{0, \ldots, k\}^n$ whose support set $\text{supp}(\mathbf{s}) \in \mathcal{I}_\mathcal{M}$, i.e., $\mathcal{C}_\mathcal{M} := \{\mathbf{s} \in \{0, \ldots, k\}^n : \text{supp}(\mathbf{s}) \in \mathcal{I}_\mathcal{M}\}$.*

**Definition 2.2** (**Knapsack constraint for $k$-submodular maximization**). *Given a non-negative vector $\mathsf{a} \in \mathbb{R}_{\geq 0}^n$, define $\mathcal{I}_\mathcal{K} := \{S : \mathsf{a}^\top 1_S \leq 1\}$, where $1_S$ is the indicator vector of set $S \subseteq [n]$. A knapsack constraint is defined to be the collection of all vectors $\mathbf{s} \in \{0, \ldots, k\}^n$ whose support set $\text{supp}(\mathbf{s}) \in \mathcal{I}_\mathcal{K}$, i.e., $\mathcal{C}_\mathcal{K} := \{\mathbf{s} \in \{0, \ldots, k\}^n : \text{supp}(\mathbf{s}) \in \mathcal{I}_\mathcal{K}\}$.*

We are now ready to define the following problem.

**Problem 2.3** (**$k$-submodular maximization with matroid and knapsack constraints**). *Given a $k$-submodular function $f : \{0, 1, \ldots, k\}^n \to \mathbb{R}_{\geq 0}$, $b$ matroid constraints $\mathcal{C}_{\mathcal{M}_1}, \ldots, \mathcal{C}_{\mathcal{M}_b}$ and $d$ knapsack constraints $\mathcal{C}_{\mathcal{K}_1}, \ldots, \mathcal{C}_{\mathcal{K}_d}$, the objective is to identify a vector $\mathbf{s} \in \{0, 1, \ldots, k\}^n$ that maximizes $f(\mathbf{s})$ subject to the constraint: $\mathbf{s} \in \left( \bigcap_{i \in [b]} \mathcal{C}_{\mathcal{M}_i} \right) \cap \left( \bigcap_{i \in [d]} \mathcal{C}_{\mathcal{K}_i} \right)$.*

Particularly, when $b = 1$ and $d = 0$, this scenario is termed $k$-submodular maximization with a matroid constraint; conversely, when $b = 0$ and $d = 1$, it is known as $k$-submodular maximization with a knapsack constraint. To solve this problem, the idea is to design continuous methods, which are widely used for the submodular case. To this end, we extend the notion of multilinear extension of submodular functions to the $k$-submodular case.

**Multilinear extension of $k$-submodular functions.**    The first step is to relax the domain of $k$-submodular functions. Recall that in the submodular context, the domain of multilinear extension is $[0, 1]^n$, which is relaxed from $\{0, 1\}^n$. For each item $i \in [n]$, a fraction value $\mathbf{x}_i \in [0, 1]$ represents a probability of selecting $i$. Now we tend to $k$-submodular functions, whose domain is $\{0, 1, \ldots, k\}^n$. The first idea is to relax this domain to $[0, k]^n$. However, it is unclear how to use a fraction value $\mathbf{x}_i \in [0, k]$ to represent a probability (distribution) in $[0, k]^n$. To address this issue, we utilize the idea of one-hot encoding that encodes a value $a \in \{0, 1, \ldots, k\}$ by a $k$-dimensional vector

$b \in \{0, 1\}^k$ with $b_j = 1$ if $a = j$ and $b_j = 0$ otherwise. This encoding motivates us to consider the domain $\Delta_k = \{\mathbf{y} \in [0, 1]^k : \sum_{j=1}^k \mathbf{y}_j \leq 1\}$ to represent probability distributions on discrete values $\{0, 1, \ldots, k\}$. To be specific, a vector $\mathbf{x} \in \Delta_k$ represents a probability distribution, where each $j \in [k]$ is assigned with probability $\mathbf{x}_j$. Thus, the domain of the $k$-multilinear extension of $k$-submodular function is defined to be the corner of the cube $\Delta_k^n := \left\{\mathbf{x} \in [0, 1]^{nk} : \sum_{j=1}^k \mathbf{x}_{i,j} \leq 1, \forall i \in [n]\right\}$. Note that $\Delta_k^n$ can be viewed as a (partition) matroid polytope with rank 1 and $nk$ elements. [2] We are ready to propose the following notion of multilinear extension of $k$-submodular functions.

**Definition 2.4** ($k$-**multilinear extension**). *Given a $k$-submodular function $f : \{0, \ldots, k\}^n \to \mathbb{R}_{\geq 0}$, the $k$-multilinear extension $F : \Delta_k^n \to \mathbb{R}_{\geq 0}$ is defined as*

$$F(\mathbf{x}) = \sum_{\mathbf{s} \in \{0,\ldots,k\}^n} f(\mathbf{s}) \prod_{i \in [n]:\mathbf{s}_i \neq 0} \mathbf{x}_{i,\mathbf{s}_i} \prod_{i \in [n]:\mathbf{s}_i = 0} \left(1 - \sum_{j=1}^k \mathbf{x}_{i,j}\right). \tag{2}$$

For every $\mathbf{x} \in \Delta_k^n$, it follows that $F(\mathbf{x}) = \mathbb{E}[f(\mathbf{s})]$ where $\mathbf{s} \in \{0, \ldots, k\}^n$ denotes a random vector: for each item $i \in [n]$, $\mathbf{s}_i = j$ for $j \in [k]$ with a probability $\mathbf{x}_{i,j}$ and otherwise, $\mathbf{s}_i = 0$, which occurs independently across all items. Multilinear extension of submodular function is a special case of Definition 2.4 when $k = 1$.

Designing continuous algorithms via multilinear extension typically necessitates computing the gradient of the function $F$. However, accurately calculating the gradient value $\nabla F(\mathbf{x})$ for a vector $\mathbf{x} \in \Delta_k^n$ involves an exponential number of queries to the value oracle $\mathcal{O}_f$, presenting a significant computational challenge. To mitigate this issue, we propose a method for constructing an approximate gradient oracle, as outlined in the following lemma. The implementation details and proofs can be found in Appendix B.

**Lemma 2.5** (**Existence of approximate oracle** $\mathcal{O}_{\nabla F}^{(\varepsilon, \eta)}$). *Given $\varepsilon, \eta \in (0, 1)$, let $F$ be the $k$-multilinear extension of $f$. There is an oracle $\mathcal{O}_{\nabla F}^{(\varepsilon, \eta)}$ that for any $\mathbf{x} \in \Delta_k^n$, calls $\mathcal{O}_f$ for at most $\left\lceil \frac{16kn^4 \log\left(\frac{n^2+1}{\varepsilon \eta}\right)}{\varepsilon^2} \right\rceil$ times and returns a stochastic estimate $\widehat{\nabla F(\mathbf{x})}$ of the gradient $\nabla F(\mathbf{x})$ such that for all $i \in [n]$ and $j \in [k]$, $\left|\widehat{\partial_{i,j} F(\mathbf{x})} - \partial_{i,j} F(\mathbf{x})\right| \leq \frac{\varepsilon M}{n\sqrt{k}}$, with probability at least $1 - \frac{\varepsilon \eta}{n^2 + 1}$.*

We find that $k$-multilinear extension enjoys several useful properties. We first note that the following *preservation of monotonicity* trivially holds: if a $k$-submodular function $f$ is monotone, its $k$-multilinear extension $F$ is also monotone.

Let $M := \max\{\max_{i,j} f(\mathbf{e}_{i,j}) - f(\mathbf{0}), 0\}$ be the maximum value of an element determined by $f$, where $\mathbf{e}_{i,j}$ is a basis vector in $\mathbb{R}^{n \times k}$ with only the $(i, j)$-th entry being 1. The following is another useful property of $F$, called *element-wise non-positive Hessian*:

$$\frac{\partial^2 F}{\partial \mathbf{x}_{i_1, j_1} \partial \mathbf{x}_{i_2, j_2}} \begin{cases} = 0 & \text{if } i_1 = i_2, \\ \in [-2M, 0] & \text{if } i_1 \neq i_2. \end{cases} \tag{3}$$

Furthermore, we observe the following *approximate linearity* property for $F$: for any points $\mathbf{x}, \mathbf{x}' \in \Delta_k^n$ satisfy that $\mathbf{x}' - \mathbf{x} \in \delta \cdot \Delta_k^n$, we have

$$F(\mathbf{x}') - F(\mathbf{x}) \geq \sum_{i \in [n], j \in [k]} (\mathbf{x}'_{i,j} - \mathbf{x}_{i,j}) \cdot \partial_{i,j} F(\mathbf{x}) - n^2 \delta^2 M. \tag{4}$$

As discussed in Section 1.2, this property is essential for the analysis of a Frank-Wolfe type algorithm. These properties, together with other useful properties, are summarized in Lemma C.1 and their proofs can be found in Appendix C.

**Continuous constraints for $k$-multilinear extension and membership oracle.** In addressing the continuous domain $\Delta_k^n$ for $k$-multilinear extension, it becomes necessary to adapt both matroid and knapsack constraints to this continuous framework. Initially, we define polytopes that correspond to these constraints:

---

[2] Despite the similarity in appearance, there is no known reduction from a $k$-submodular function to a submodular function with a partition matroid constraint. See Appendix A.3 for details.

For a matroid constraint defined by a collection $\mathcal{I}_\mathcal{M}$, we establish its corresponding polytope $\mathcal{P}_\mathcal{M}$ as $\mathrm{conv}\{1_S : S \in \mathcal{I}_\mathcal{M}\}$, where $1_S$ denotes the indicator vector of set $S$. For a knapsack constraint characterized by a vector $\mathsf{a} \in \mathbb{R}^n_{\geq 0}$, we define the corresponding polytope $\mathcal{P}_\mathcal{K}$ as $\{\mathbf{x} \in [0,1]^n : \mathsf{a}^\top \mathbf{x} \leq 1\}$. Another important notion is how a fractional vector within $\Delta^n_k$ relates to a polytope. Given a vector $\mathbf{x} \in \Delta^n_k$, we say $\mathbf{x}$ is consistent with a polytope $\mathcal{P}$, denoted as $\mathbf{x} \sim \mathcal{P}$, if $\left(\sum_{j=1}^k \mathbf{x}_{1,j}, \ldots, \sum_{j=1}^k \mathbf{x}_{n,j}\right) \in \mathcal{P}$. Additionally, we define a membership oracle $\mathcal{O}_\mathcal{P}$ that, given a vector $\mathbf{x} \in \Delta^n_k$, determines whether $\mathbf{x} \sim \mathcal{P}$.

With these definitions in place, we can illustrate how to relax matroid and knapsack constraints for vectors in $\Delta^n_k$: A vector $\mathbf{x}$ satisfies a matroid constraint if $\mathbf{x} \sim \mathcal{P}_\mathcal{M}$, and it satisfies a knapsack constraint if $\mathbf{x} \sim \mathcal{P}_\mathcal{K}$. We further present a lemma demonstrating the existence of a membership oracle for these relaxed constraints.

**Lemma 2.6** (**Existence of membership oracle (Cunningham, 1984)**). *For any constraint $\mathcal{P} = \left(\bigcap_{i \in [b]} \mathcal{P}_{\mathcal{M}_b}\right) \cap \left(\bigcap_{i \in [d]} \mathcal{P}_{\mathcal{K}_d}\right)$ of the intersection of $b$ matroid constraints and $d$ knapsack constraints, there exists an efficient membership oracle $\mathcal{O}_\mathcal{P}$.*

It is well-established that with access to a membership oracle $\mathcal{O}_\mathcal{P}$, linear optimization problems involving the function $F$ can be efficiently addressed using Frank-Wolfe-type methods, as noted by Lee et al. (2018). To address Problem 2.3, we frame a constrained optimization challenge within a continuous domain: We aim to maximize a $k$-multilinear extension $F : \Delta^n_k \to \mathbb{R}_{\geq 0}$ at a fractional point $\mathbf{x} \in \Delta^n_k$, subject to the constraint $\mathbf{x} \sim \mathcal{P}$, where $\mathcal{P}$ is a down-closed convex polytope.

Define $\mathbf{o}^\star := \arg\max_{\mathbf{x} \sim \mathcal{P}} F(\mathbf{x})$ as the optimal fractional solution for this continuous optimization scenario. We assume that for every $i \in [n]$, the unit vector $e_i$ is within $\mathcal{P}$. If not, the $i$-th element is irrelevant to the constrained $k$-submodular maximization problem and can be excluded. With this premise, it follows that $F(\mathbf{o}^\star) \geq F(\mathbf{e}_{i,j}) = f(\mathbf{e}_{i,j})$. Recalling that $M = \max\{\max_{i,j} f(\mathbf{e}_{i,j}) - f(\mathbf{0}), 0\}$, we establish that $F(\mathbf{o}^\star) \geq M$.

## 3  RESULTS FOR MONOTONE $k$-SUBMODULAR MAXIMIZATION

In this section, we consider the case that the objective $k$-submodular function $f$ is monotone.

**Theorem 3.1** (**Main theorem I, monotone case**). *There exists a polynomial-time algorithm that given a monotone $k$-submodular $f : \{0, 1, \ldots, k\}^n$ and a constraint polytope $\mathcal{P} \subseteq \Delta^n_k$, with probability at least $1 - \eta$, outputs a solution that is*

- *$(\frac{1}{2} - \varepsilon)$-approximate under a single matroid constraint, with calling $\mathcal{O}_f$ at most $O\left(\frac{kn^6 \log\left(\frac{n}{\varepsilon\eta}\right)}{\varepsilon^3}\right)$ times and calling $\mathcal{O}_\mathcal{P}$ at most $O\left(\frac{k^3 n^6 \log\left(\frac{n}{\varepsilon\eta}\right)}{\varepsilon^2}\right)$, for any fixed $\varepsilon > 0$;*

- *$(\frac{1}{2} - \varepsilon)$-approximate under the intersection of $O(1)$ knapsack constraints, with calling $\mathcal{O}_f$ at most $O\left(k^{poly(\frac{1}{\varepsilon})} n^{poly(\frac{1}{\varepsilon})}\right)$ times and calling $\mathcal{O}_\mathcal{P}$ at most $O\left(k^{poly(\frac{1}{\varepsilon})} n^{poly(\frac{1}{\varepsilon})}\right)$, for any fixed $\varepsilon > 0$;*

- *$(\frac{0.3}{b} - \varepsilon)$-approximate under the intersection of $b$ matroid constraints and $O(1)$ knapsack constraints, with calling $\mathcal{O}_f$ at most $O\left(k^{poly(\frac{1}{\varepsilon})} n^{poly(\frac{1}{\varepsilon})}\right)$ times and calling $\mathcal{O}_\mathcal{P}$ at most $O\left(k^{poly(\frac{1}{\varepsilon})} n^{poly(\frac{1}{\varepsilon})}\right)$, for any fixed $\varepsilon > 0$.*

The difference in approximation guarantees and query complexity arises due to the rounding procedure's impact under varying constraints. Our algorithm maintains a consistent approximation ratio for the $k$-multilinear extension across down-closed constraints (Lemma 3.3), but the rounding effectiveness varies, leading to different guarantees (Lemma 3.2).

We remark that our query complexity is usually larger than existing combinatorial methods, e.g., a 0.432-approximate algorithm with a query complexity of $O(k^9 n^{10})$ for a single knapsack constraint (Niu et al., 2023). Nevertheless, the focus of this paper is to improve the approximation ratio, specifically achieving (asymptotically) optimal approximation algorithms.

In Section 3.1, we propose a unified optimization framework thereof, and in Section 3.2, we analyze the approximation ratio in Theorem 3.1. The complete proof of Theorem 3.1 can be found in Appendix E. We can also extend the result to non-monotone case; see Appendix F.

## 3.1 THE ALGORITHM

In this section, we propose the algorithm for Theorem 3.1 using $k$-multilinear extension. The algorithm consists of two stages: a continuous optimization stage and a rounding stage.

**The continuous optimization stage.** In the first stage, we design a Frank-Wolfe-type method to approximately maximize the $k$-multilinear extension (Algorithm 1).

---

**Algorithm 1:** Frank-Wolfe algorithm for the monotone case, $\texttt{FW}(f, \mathcal{P}, \varepsilon, \eta)$

---

**Input :** $\mathcal{O}_f, \mathcal{O}_\mathcal{P}$ and hyperparameters $\varepsilon, \eta \in (0, 1)$.

1   **Initialize:** $\mathbf{x}(0) \leftarrow \mathbf{0}$, $t \leftarrow 0$; stepsize $\delta = \frac{1}{N}$ with $N = \lceil \frac{n^2}{\varepsilon} \rceil$, $\mathcal{O}_{\nabla F}^{(\varepsilon, \eta)}$ by Lemma 2.5.

2   **while** $t < 1$ **do**

3      find a direction $\mathbf{v}(t) = \arg\max_{\mathbf{v} \in \Delta_k^n, \mathbf{v} \sim \mathcal{P}} \langle \widehat{\nabla F(\mathbf{x}(t))}, \mathbf{v} \rangle$          ▷ By LP

4      $\mathbf{x}(t + \delta) = \mathbf{x}(t) + \delta \mathbf{v}(t)$, $t \leftarrow t + \delta$

5   **return** $\mathbf{x}(\mathbf{1})$.

---

The Frank-Wolfe algorithm terminates after the $N$-th iteration. During each iteration, the surrogate function $\langle \nabla F(\mathbf{s}(t)), \mathbf{v}(t) \rangle$ is utilized to identify the feasible direction that maximizes the improvement in the function value. This process involves maximizing a linear objective subject to constraints defined within a polytope in the positive coordinate space. The computational effort required per iteration is comparable to solving a positive linear program (LP), for which a quadratic time solver is documented in Lee et al. (2018).

**The Rounding Stage.** Let $\mathbf{x}(\mathbf{1})$ represent the fractional solution obtained from Algorithm 1. The subsequent stage involves rounding $\mathbf{x}(\mathbf{1})$ to an integral solution. The performance of this rounding process is encapsulated in the following lemma.

**Lemma 3.2 (Rounding scheme).** *Let $\epsilon, \eta \in (0, 1)$, and $\mathcal{P}$ be a polytope. Suppose for any monotone $k$-submodular function $f'$, Algorithm $\texttt{FW}(f', \mathcal{P}, \varepsilon, \eta)$ outputs a solution $\mathbf{x} \in \Delta_k^n$ that satisfies $F(\mathbf{x}) \geq \alpha \max_{\mathbf{y} \in \Delta_k^n, \mathbf{y} \sim \mathcal{P}} F(\mathbf{y})$. Then for any $\varepsilon > 0$, there exists a rounding scheme that outputs a solution $\mathbf{s} \in \{0, \ldots, k\}^n$ with $\mathbf{s} \sim \mathcal{P}$, that is*

- *$\alpha$-approximate under a single matroid constraint, i.e., $f(\mathbf{s}) \geq \alpha \cdot \max_{\mathbf{s}' \in \{0, \ldots, k\}^n, \mathbf{s}' \sim \mathcal{P}} f(\mathbf{s}')$, with calling $\texttt{FW}$ one time and calling $\mathcal{O}_\mathcal{P}$ at most $O(Nn^2)$ times;*

- *$\alpha(1 - \varepsilon)$-approximate under $O(1)$ knapsack constraints, with calling $\texttt{FW}, O_f, \mathcal{O}_\mathcal{P}$ at most $O\left(k^{poly(1/\varepsilon)} n^{poly(1/\varepsilon)}\right)$ times;*

- *$\left(\frac{0.6\alpha}{b}(1 - \varepsilon)\right)$-approximate under the intersection of $b$ matroid constraints and $l = O(1)$ knapsack constraints, with calling $\texttt{FW}, O_f, \mathcal{O}_\mathcal{P}$ at most $O\left(k^{poly(1/\varepsilon)} n^{poly(1/\varepsilon)}\right)$ times.*

Our rounding scheme is an extension of the approaches developed for submodular maximization, as detailed in works by (Călinescu et al., 2011; Chekuri et al., 2014; 2010b). Specifically, for a single matroid constraint, the rounding procedure is directly applied to the output of $\texttt{FW}(f, \mathcal{P}, \epsilon, \eta)$, necessitating only a single invocation of the $\texttt{FW}$ algorithm. For knapsack constraints, where elements might exhibit large costs or significantly influence the function value, an enumeration stage akin to that used in submodular maximization (Chekuri et al., 2014) is employed, resulting in a complexity of $O(n^{poly(1/\varepsilon)})$. This process is extended by first employing the continuous maximization algorithm $\texttt{FW}(f', \mathcal{P}', \epsilon, \eta)$, where $f'$ and $\mathcal{P}'$ represent a marginal function and a restricted polytope specifically for large elements. By enumerating all possible restrictions and selecting the optimal result, the complexity becomes $O(k^{poly(1/\varepsilon)} n^{poly(1/\varepsilon)})$, given that the number of large elements is capped at $O(poly(1/\varepsilon))$. Each large element has $k$ potential assignments, amplifying the complexity by a factor of $O(k^{poly(1/\varepsilon)})$. Details and proofs are provided in Appendix D.

## 3.2 Performance analysis of Algorithm 1

We focus on proving the approximation ratio of Theorem 3.1 in the main body. Recall that $\mathbf{o}^\star := \arg\max_{\mathbf{x}\in\Delta_k^n, \mathbf{x}\sim\mathcal{P}} F(\mathbf{x})$. By Lemma 3.3, it suffices to prove the following key lemma.

**Lemma 3.3 (Analysis of the Frank-Wolfe algorithm).** *When $f$ is monotone, then $F(\mathbf{x}(1)) \geq \left(\frac{1}{2} - 2\varepsilon\right) F(\mathbf{o}^\star)$, with probability at least $1 - \eta$.*

**Technical novelty in the proof of Lemma 3.3.** As discussed in Section 1.2, due to the closure challenge, we consider the following operation of linear combination: for each time step $t \in [0, 1]$, we generate an auxiliary point $\mathbf{o}(t) = \mathbf{x}(t) + (1-t)\mathbf{o}^\star$. Since $\mathbf{x}(t) \in t \cdot \Delta_k^n$, this auxiliary point $\mathbf{o}(t)$ must be within $\Delta_k^n$. We find that, to prove Lemma 3.3, it suffices to prove

$$F(\mathbf{x}(t+\delta)) - F(\mathbf{x}(t)) \geq F(\mathbf{o}(t)) - F(\mathbf{o}(t+\delta)) - 3\varepsilon M\delta, \tag{5}$$

which lower-bounds the value gain of each iteration. Recall that $M = \max\{\max_{i,j} f(\mathbf{e}_{i,j}) - f(\mathbf{0}), 0\}$. To see this, we note that by summing over all $t$,

$$F(\mathbf{x}(1)) \geq F(\mathbf{x}(1)) - F(\mathbf{x}(0)) \geq F(\mathbf{o}(0)) - F(\mathbf{o}(1)) - 3\varepsilon M = F(\mathbf{o}^\star) - F(\mathbf{x}(1)) - 3\varepsilon M.$$

Rearranging this inequality, we obtain that $F(\mathbf{x}(1)) \geq \frac{1}{2}(\mathbf{o}^\star) - 3\varepsilon M$. Recall that $F(\mathbf{o}^\star) \geq M$. Thus, we conclude that $F(\mathbf{x}(1)) \geq \left(\frac{1}{2} - 2\varepsilon\right) F(\mathbf{o}^\star)$.

It remains to prove Ineq. (5). The key is to provide a lower bound for $F(\mathbf{x}(t+\delta)) - F(\mathbf{x}(t))$ and an upper bound for $F(\mathbf{o}(t)) - F(\mathbf{o}(t+\delta))$. We first note that $F(\mathbf{x}(t+\delta)) - F(\mathbf{x}(t)) \geq \langle \nabla F(\mathbf{x}(t)), \mathbf{o}^\star \rangle \delta - 3\varepsilon M\delta$. To see this, using Ineq. (4) and the fact that $\delta \leq \frac{\varepsilon}{n^2}$, we have

$$F(\mathbf{x}(t+\delta)) - F(\mathbf{x}(t)) \geq \langle \nabla F(\mathbf{x}(t)), \mathbf{v}(t) \rangle \delta - n^2\delta^2 M \geq \langle \nabla F(\mathbf{x}(t)), \mathbf{v}(t) \rangle \delta - \varepsilon M\delta, \tag{6}$$

where $\mathbf{v}(t)$ is the local optimal direction obtained in Line 3 of Algorithm 1. Furthermore, using the theoretical guarantee of approximate oracle $\mathcal{O}_{\nabla F}^{(\varepsilon,\eta)}$ stated in Lemma 2.5, we can conclude that $\langle \nabla F(\mathbf{x}(t)), \mathbf{v}(t) \geq \langle \nabla F(\mathbf{x}(t)), \mathbf{o}^\star \rangle - 2\varepsilon M$. Combining with Ineq. (6), we can conclude that $F(\mathbf{x}(t+\delta)) - F(\mathbf{x}(t)) \geq \langle \nabla F(\mathbf{x}(t)), \mathbf{o}^\star \rangle \delta - 3\varepsilon M\delta$.

Next, we show that $F(\mathbf{o}(t)) - F(\mathbf{o}(t+\delta)) \leq \langle \nabla F(\mathbf{x}(t)), \mathbf{o}^\star \rangle \delta$. This inequality, when combined with Ineq. (6), concludes Ineq. (5). To derive this bound, we define the component-wise minimum of $\mathbf{o}(t)$ and $\mathbf{o}(t+\delta)$ as $\mathbf{o}'(t) = \mathbf{x}(t) + (1 - t - \delta)\mathbf{o}^\star$. By analyzing the trajectory from $\mathbf{o}(t+\delta)$ to $\mathbf{o}'(t)$ and then to $\mathbf{o}(t)$, we observe a decrease in each coordinate during the first transition and an increase during the second. Given the function's monotonicity, we infer that $F(\mathbf{o}(t)) - F(\mathbf{o}(t+\delta)) \leq F(\mathbf{o}(t)) - F(\mathbf{o}')$. Furthermore, leveraging the concavity along the positive direction $\mathbf{o}(t) - \mathbf{o}' = \delta\mathbf{o}^\star$ as specified by Eq. (3), we deduce that $F(\mathbf{o}(t)) - F(\mathbf{o}') \leq \langle \nabla F(\mathbf{o}'(t)), \mathbf{o}^\star \rangle \delta$. Finally, considering the non-increasing nature of the partial derivative (also confirmed by Eq. (3) and the fact that $\mathbf{o}^\star \in \Delta_n^k$, we establish that $\langle \nabla F(\mathbf{o}'(t)), \mathbf{o}^\star \rangle \delta \leq \langle \nabla F(\mathbf{x}(t)), \mathbf{o}^\star \rangle \delta$. Thus, summarizing the above findings, we confirm that $F(\mathbf{o}(t)) - F(\mathbf{o}(t+\delta)) \leq \langle \nabla F(\mathbf{x}(t)), \mathbf{o}^\star \rangle \delta$.

The details of the proof of Lemma 3.3 can be found in Appendix E.

## 4 Conclusions and future works

We introduce a unified Frank-Wolfe-type framework for addressing $k$-submodular maximization across various settings. Notably, we achieved an optimal $1/2$-approximation for monotone $k$-submodular functions and a $1/3$-approximation for non-monotone functions under constraints of a single matroid and $O(1)$ knapsacks. Our framework is adaptable to a broad range of constraints, including any combination of matroid and knapsack constraints.

The foundation of our frameworks is the multilinear extension of $k$-submodular functions, which facilitates the design of maximization algorithms that allow for flexible step sizes and update directions. Given the success of multilinear extensions in achieving optimal outcomes in numerous submodular maximization scenarios, our approach to extending and rounding $k$-submodular functions may present novel avenues for further research and application.

There are several interesting directions for future exploration. Determining tight approximation ratios for non-monotone $k$-submodular maximization is an intriguing challenge that continues to be of significant academic interest. Furthermore, the potential to derandomize $k$-submodular maximization algorithms offers a valuable area of inquiry, particularly for applications that benefit from deterministic outputs, enhancing both reproducibility and consistency in practical deployments.

ACKNOWLEDGMENTS

HZ was supported by International Graduate Program of Innovation for Intelligent World and Next Generation Artificial Intelligence Research Center. LH has been supported by the New Cornerstone Science Foundation. BW was partially supported by the National Natural Science Foundation of China (62106213, 72394361), Longgang District Key Laboratory of Intelligent Digital Economy Security, and an extended support project from the Shenzhen Science and Technology Program.

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

CONTENTS

# A ADDITIONAL DISCUSSION

## A.1 MORE EXAMPLES OF $k$-SUBMODULAR APPLICATION IN MACHINE LEARNING

While submodular maximization may be more famous in the ML community, many applications of it could be extended to $k$-submodular maximization. One example is diversity, where the selection needs to balance multiple sources.

- **Feature selection**: In machine learning, feature selection is the process of identifying a subset of features that are most relevant to a given task. $k$-submodular maximization can be used to find a diverse set of features that maximizes the performance of a model.
- **Active learning**: Active learning is a technique for selecting the most informative data points to label, which can help to reduce the cost of labeling data. $k$-submodular maximization can be used to select a diverse set of data points that are likely to provide the most information about the underlying model.
- **Recommendation systems**: Recommendation systems are used to provide personalized recommendations to users. $k$-submodular maximization can be used to select a diverse set of items that are likely to be of interest to a given user.

There are many other applications. When we mention sensor placement in Section 1, this is also related to data acquisition in machine learning. Determining the optimal placement of sensors to collect the most informative data for training a model. It is also useful in anomaly detection, where one trategically places monitoring agents within a network to maximize the chances of detecting anomalies. Meanwhile, $k$-submodular optimization is useful for resource allocation tasks, which is relevant in several ML senarios. In distributed computing, one may assign tasks to a limited number of computing nodes to optimize ML training performance and energy consumption. In cloud ML, one may allocate different types of virtual machines or containers to meet varying workloads while minimizing costs.

## A.2 AN EQUIVALENT DEFINITION OF $k$-SUBMODULAR FUNCTIONS

In this subsection, we demonstrate the equivalence between the definition of $k$-submodularity (Eq. (1)) and another definition (Definition A.1) in the literature. We first recall another definition as follows.

**Definition A.1** (Alterative definition of $k$-submodular Sakaue (2017)). *Let* $(k+1)^n :=\{(X_1,\ldots,X_k) \mid X_i \subseteq [n]\,(i=1,\ldots,k),\,X_i \cap X_j = \emptyset\,(i \neq j)\}$. *Then, a function* $f : (k+1)^n \to \mathbb{R}$ *is called $k$-submodular if, for any* $\boldsymbol{X} = (X_1,\ldots,X_k)$ *and* $\boldsymbol{Y} = (Y_1,\ldots,Y_k)$ *in* $(k+1)^n$, *we have*

$$f(\boldsymbol{X}) + f(\boldsymbol{Y}) \geq f(\boldsymbol{X} \sqcup \boldsymbol{Y}) + f(\boldsymbol{X} \sqcap \boldsymbol{Y}) \tag{7}$$

*where*

$$\boldsymbol{X} \sqcap \boldsymbol{Y} := (X_1 \cap Y_1,\ldots,X_k \cap Y_k),$$

$$\boldsymbol{X} \sqcup \boldsymbol{Y} := \left(X_1 \cup Y_1 \backslash\Big(\bigcup_{i\neq 1} X_i \cup Y_i\Big),\ldots,X_k \cup Y_k \backslash\Big(\bigcup_{i\neq k} X_i \cup Y_i\Big)\right).$$

We also recall our definition defined by Eq. (1) in Section 1.

**Definition A.2** (Our definition of $k$-submodular). *For an integer $k \geq 1$ and a finite nonempty set* $[n]$, *a non-negative function* $f : \{0,\ldots,k\}^n \to \mathbb{R}_{\geq 0}$ *is called $k$-submodular if for all* $\mathbf{s}$ *and* $\mathbf{t}$ *in* $\{0,\ldots,k\}^n$, *we have*

$$f(\mathbf{s}) + f(\mathbf{t}) \geq f(\min_0(\mathbf{s},\mathbf{t})) + f(\max_0(\mathbf{s},\mathbf{t})), \tag{8}$$

*where for every* $i \in [n]$,

$$\min_0(\mathbf{s},\mathbf{t})_i = \begin{cases} 0, & \mathbf{s}_i\mathbf{t}_i \neq 0 \text{ and } \mathbf{s}_i \neq \mathbf{t}_i, \\ \min(\mathbf{s}_i,\mathbf{t}_i), & \text{otherwise,} \end{cases} \qquad \text{and}$$

$$\max_0(\mathbf{s},\mathbf{t})_i = \begin{cases} 0, & \mathbf{s}_i\mathbf{t}_i \neq 0 \text{ and } \mathbf{s}_i \neq \mathbf{t}_i, \\ \max(\mathbf{s}_i,\mathbf{t}_i), & \text{otherwise.} \end{cases}$$

We now show the equivalence of these two definitions.

**Equivalence of $\{0, \ldots, k\}^n$ and $(k+1)^n$.** It is straightforward to verify that the two sets are equivalent: an element $e$ belongs to $S_i$ if and only if $\mathbf{s}_e = i$. Furthermore, $\mathbf{s}_e = 0$ if and only if $e$ from the set $[n]$ is not included in any of the sets $S_1, \ldots, S_k$.

**Equivalence of Eq. (7) and Eq. (8).** The equality of the left side is clear. For the right part, the following claim is made:

- **Intersection** ($\sqcap$ in Definition A.1 and $\min_0$ in Definition A.2):
    - For Definition A.1, $(\boldsymbol{X} \sqcap \boldsymbol{Y})_i = X_i \cap Y_i$.
    - For Definition A.2, $(\min_0(\mathbf{s}, \mathbf{t}))_i = \min(\mathbf{s}_i, \mathbf{t}_i)$ when $s_i, t_i \in \{0, i\}$, aligning with the intersection of sets for corresponding labels. If $\mathbf{s}_i$ and $\mathbf{t}_i$ are different and nonzero, the result is $0$, representing the empty intersection.
- **Union** ($\sqcup$ in Definition A.1 and $\max_0$ in Definition A.2):
    - For Definition A.1, $(\boldsymbol{X} \sqcup \boldsymbol{Y})_i = X_i \cup Y_i \setminus \bigcup_{j \neq i}(X_j \cup Y_j)$, which ensures disjointness.
    - For Definition A.2, $(\max_0(\mathbf{s}, \mathbf{t}))_i = \max(\mathbf{s}_i, \mathbf{t}_i)$ when $\mathbf{s}_i, \mathbf{t}_i \in \{0, i\}$, aligning with the union of sets. If $\mathbf{s}_i$ and $\mathbf{t}_i$ are different and nonzero, the result is $0$, ensuring disjointness.

Both definitions use the inequality:

$$f(x) + f(y) \geq f(x \cup y) + f(x \cap y),$$

which holds in both formulations because the operations $\sqcup, \sqcap$ in Definition A.1 are equivalent to $\max_0, \min_0$ in Definition A.2, and the domains and function mappings are equivalent.

### A.3 $k$-SUBMODULAR CAN NOT BE REDUCED TO SUBMODULAR WITH A PARTITION MATROID

In this section, we revisit the findings of Singh et al. (2012a), which demonstrate that non-negative 2-submodular functions (i.e., $k = 2$) cannot be universally reduced to general non-negative submodular functions within a partition matroid framework. Below we explain the reason.

One possible reduction is as follows. We define the domain as $\bar{\Delta}_k^n \subseteq \{0, 1\}^{nk}$ where $\bar{\Delta}_k = \{x \in \{0, 1\}^k : \sum_{j=1}^k x_j \leq 1\}$, and define a function $\bar{f} : \bar{\Delta}_k^n \to \mathbb{R}$ as $\bar{f}(S) = f(\mathbf{s})$, where $\mathbf{s}$ is defined as $\mathbf{s}_i = j$ if there exists a unique element $e_{i,j} \in S$ and $\mathbf{s}_i = 0$ otherwise. However, we may not be able to extend the domain of such a submodular function to $\{0, 1\}^{nk}$ without violating non-negativity or monotonicity, even for the simplest case of $k = 2$. Specifically:

- As shown in Lemma 2 of Singh et al. (2012a), there exists a non-negative 2-submodular function, for which no extension is both non-negative and submodular.
- Furthermore, Lemma 3 of Singh et al. (2012a) demonstrates that there exists a monotone, non-negative 2-submodular function, for which no extension is non-negative, monotone, and submodular.

## B   PROOF OF LEMMA 2.5: EXISTENCE OF AN EFFICIENT ORACLE $\mathcal{O}_F$

We recall Lemma 2.5 states that there exists a (stochastic) gradient oracle $\mathcal{O}_{\nabla F}^{(\varepsilon, \eta)}$ for $\nabla F$ with parameters $\varepsilon, \delta \in (0, 1)$ where for any query $\mathbf{x} \in \Delta_k^n$, $\mathcal{O}_{\nabla F}^{(\varepsilon, \eta)}$ provides a stochastic estimate $\widehat{\nabla F(\mathbf{x})}$ that is "$\frac{\varepsilon M}{kn^2}$-close" to the gradient $\nabla F(\mathbf{x})$ in terms of $\ell_\infty$-norms, with a probability at least $1 - \eta$.

**Lemma B.1 (Existence of oracle $\mathcal{O}_{\nabla F}^{(\varepsilon, \eta)}$).** *Given $\varepsilon, \eta \in (0, 1)$, let $F$ be the $k$-multilinear extension of $f$. There is an algorithm that for any point $\mathbf{x} \in \Delta_k^n$, calls $\mathcal{O}_f$ for at most $\lceil \frac{16kn^4 \log\left(\frac{n^2+1}{\varepsilon \eta}\right)}{\varepsilon^2} \rceil$ times and returns a stochastic estimate $\widehat{\nabla F(\mathbf{x})}$ of the gradient $\nabla F(\mathbf{x})$ such that for all $i \in [n]$ and $j \in [k]$,*

$$\left| \widehat{\partial_{i,j} F(\mathbf{x})} - \partial_{i,j} F(\mathbf{x}) \right| \leq \frac{\varepsilon M}{n\sqrt{k}},$$

with probability at least $1 - \frac{\varepsilon\eta}{n^2+1}$.

*Proof.* Given oracle access to a $k$-submodular function $f$, the Chernoff bounds (see Theorem A.1.16 in (Alon & Spencer, 2008)) implies the following theorem which allows us to approximate the value of the $k$-multilinear extension $F$ to arbitrary accuracy.

**Lemma B.2.** *Assume $F$ is the $k$-multilinear extension of $f$. Given a point $\mathbf{x} \in \Delta_k^n$, if $\mathbf{s}^1, \ldots, \mathbf{s}^t \in \{0, \ldots, k\}^n$ are random vectors independently sampled as follows: for each $l \in [t]$, for each item $i \in [n]$, $\mathbf{s}_i^l = j$ for $j \in [k]$ with probability $\mathbf{x}_{i,j}$ and otherwise, $\mathbf{s}_i^l = 0$, which occurs independently across all items; then for any $\varepsilon_0 \in (0, 1)$, we have*

$$\left| \frac{1}{t} \sum_{i=1}^{t} f(\mathbf{s}^i) - F(\mathbf{x}) \right| \leq \varepsilon_0 | \max_{\mathbf{s} \in \Delta_k^n} f(\mathbf{s}) |$$

*with probability at least $1 - e^{-t\varepsilon_0^2/4}$.*

For any partial derivative $\partial_{i,j} F(\mathbf{x})$ at point $\mathbf{x} \in \Delta_k^n$ and direction $\mathbf{e}_{i,j}$, we construct its stochastic estimate $\widehat{\partial_{i,j} F(\mathbf{x})}$ as follows. Consider points $\mathbf{x}^0, \mathbf{x}^1 \in \Delta_k^n$ defined as

$$\mathbf{x}_{p,q}^0 = \begin{cases} 0 & \text{If } p = i, \\ \mathbf{x}_{p,q} & \text{Otherwise.} \end{cases} \quad \text{and} \quad \mathbf{x}_{p,q}^1 = \begin{cases} 0 & \text{If } p = i, q \neq j, \\ 1 & \text{If } p = i, q = j, \\ \mathbf{x}_{p,q} & \text{Otherwise.} \end{cases}$$

We observe that the Hessian elements of $F$ satisfy the condition $\frac{\partial^2 F}{\partial x_{i,j_1} \partial x_{i,j_2}} = 0$, for all $i \in [n]$ and $j_1, j_2 \in [k]$. This implies that $\partial_{i,j} F(\mathbf{x}^0) = \partial_{i,j} F(\mathbf{x})$. Leveraging the multilinearity of $F$, we deduce that

$$F(\mathbf{x}^1) - F(\mathbf{x}^0) = \partial_{i,j} F(\mathbf{x}^0) = \partial_{i,j} F(\mathbf{x}).$$

We consider two sets of independent samples of random vectors, $\mathbf{s}^{0,1}, \ldots, \mathbf{s}^{0,t}$ and $\mathbf{s}^{1,1}, \ldots, \mathbf{s}^{1,t}$, which satisfy the property delineated in Lemma B.2 for the points $\mathbf{x}^0$ and $\mathbf{x}^1$, respectively. Define $\widehat{\partial_{i,j} F(\mathbf{x})} = \frac{1}{t} \sum_{i=1}^{t} f(\mathbf{s}_i^1) - \frac{1}{t} \sum_{i=1}^{t} f(\mathbf{s}_i^0)$. Then by Lemma B.2 the concentration property holds as

$$\begin{aligned} \left| \widehat{\partial_{i,j} F(\mathbf{x})} - \partial_{i,j} F(\mathbf{x}) \right| &= \left| \frac{1}{t} \sum_{i=1}^{t} f(\mathbf{s}_i^1) - F(\mathbf{x}^1) - \left( \frac{1}{t} \sum_{i=1}^{t} f(\mathbf{s}_i^0) - F(\mathbf{x}^0) \right) \right| \\ &\leq \left| \frac{1}{t} \sum_{i=1}^{t} f(\mathbf{s}_i^1) - F(\mathbf{x}^1) \right| + \left| \frac{1}{t} \sum_{i=1}^{t} f(\mathbf{s}_i^0) - F(\mathbf{x}^0) \right| \\ &\leq 2\varepsilon_0 | \max_{\mathbf{s} \in \Delta_k^n} f(\mathbf{s}) | \\ &\leq 2\varepsilon_0 n M, \end{aligned}$$

with probability at least $1 - 2e^{-t\varepsilon_0^2/4}$. By setting $\varepsilon_0 = \frac{\varepsilon}{2k^{1/2}n^2}$ and $t = \lceil \frac{16kn^4 \log\left(\frac{n^2+1}{\varepsilon\eta}\right)}{\varepsilon^2} \rceil$ we prove the lemma. $\square$

As a direct corollary, we know that $\left\| \widehat{\nabla F(\mathbf{x})} - \nabla F(\mathbf{x}) \right\|_2 \leq \frac{\varepsilon M}{\sqrt{n}}$ holds for any point $\mathbf{x} \in \Delta_k^n$, which is useful for our analysis.

For the general case of a $k$-submodular function $f$, computing the $k$-multilinear extension $F$ poses significant challenges. This computational challenge also appears for submodular functions ($k = 1$). Nevertheless, we have identified instances, such as the MAX-$k$-CUT problem, where the $k$-multilinear extension $F$ and its gradient $\nabla F$ can be explicitly calculated. This extends the multilinear extension for the MAX-CUT problem, which is shown to have an efficient gradient oracle (Chen & Kuhnle, 2024, Appendix G).

Given a weighted undirected graph $G = (V, E)$ together with a weight function $w$ on edges, the goal of the MAX-$k$-CUT problem is to partition the vertices into $k$ distinct parts, such that the

total weight of the edges across the parts is maximized, i.e., maximizing the cut value of a partition $\mathbf{s} \in \{0, 1, \ldots, k-1\}^n$, defined as

$$f(\mathbf{s}) = \sum_{(u,v) \in E} w_{u,v} \cdot 1(\mathbf{s}_u \neq \mathbf{s}_v).$$

By Iwata et al. (2016), this function $f$ is a $k$-submodular function. In this case, we can verify that its $k$-multilinear extension is of the form:

$$F(\mathbf{x}) = \sum_{(u,v) \in E} w_{u,v} \cdot (1 - (1 - \sum_{j=1}^{k-1} \mathbf{x}_{u,j})(1 - \sum_{j=1}^{k-1} \mathbf{x}_{v,j}) - \sum_{j=1}^{k-1} \mathbf{x}_{u,j}\mathbf{x}_{v,j}),$$

and its gradient is of the form:

$$\nabla_{u,j} F(\mathbf{x}) = \sum_{(u,v) \in E} w_{u,v} \cdot (1 - \sum_{j=1}^{k-1} \mathbf{x}_{v,j} - \mathbf{x}_{v,j}).$$

The computation time for $F(\mathbf{x})$ and its gradient $\nabla F(\mathbf{x})$ is $O(n^2 k)$, which is efficient. For the multilinear extension of submodular maximization, a similar argument for MAX-CUT can be found in Appendix G of Chen & Kuhnle (2024).

## C  PROPERTIES OF $k$-MULTILINEAR EXTENSION

The following lemma presents good properties for $k$-multilinear extension, which are useful for algorithm design.

**Lemma C.1** (**Properties of $k$-multilinear extension**). *Let $f \colon \{0, \ldots, k\}^n \to \mathbb{R}_{\geq 0}$ be a $k$-submodular function. Then its multilinear extension $F \colon \Delta_k^n \to \mathbb{R}_{\geq 0}$ satisfies the following properties:*

- *(Preservation of monotonicity) If $f$ is monotone, $F$ is monotone, i.e., for any point $\mathbf{x} \in \Delta_k^n$, $\partial_{i,j} F(\mathbf{x}) \geq 0$ for all $i \in [n]$ and $j \in [k]$.*

- *(Pairwise monotonicity) For all $i \in [n], j_1, j_2 \in [k]$, $\frac{\partial F}{\partial \mathbf{x}_{i,j_1}} + \frac{\partial F}{\partial \mathbf{x}_{i,j_2}} \geq 0$, i.e., along any direction $e_{i,j_1} + e_{i,j_2}$, the $k$-multilinear extension is non-decreasing.*

- *(Multilinearity) For every $i \in [n], j \in [k]$ and $\mathbf{x}, \mathbf{x}' \in \Delta_k^n$ with $\mathbf{x}' - \mathbf{x} = c \cdot e_{i,j}$,[3] the equality $\partial_{i,j} F(\mathbf{x}) = \partial_{i,j} F(\mathbf{x}')$ holds, i.e., along every coordinate direction, the corresponding directional derivative remains constant.*

- *(Element-wise non-positive Hessian) Let $M := \max\{\max_{i,j} F(\mathbf{e}_{i,j}) - F(\mathbf{0}), 0\}$ be a value determined by $f$. For all $i_1, i_2 \in [n], j_1, j_2 \in [k]$,*

$$\frac{\partial^2 F}{\partial \mathbf{x}_{i_1,j_1} \partial \mathbf{x}_{i_2,j_2}} \begin{cases} = 0 & \text{if } i_1 = i_2, \\ \in [-2M, 0] & \text{if } i_1 \neq i_2. \end{cases}$$

- *(Approximate linearity) For any points $\mathbf{x}, \mathbf{x}' \in \Delta_k^n$ satisfy that $\mathbf{x}' - \mathbf{x} \in \delta \cdot \Delta_k^n$, then*

$$F(\mathbf{x}') - F(\mathbf{x}) \geq \sum_{i \in [n], j \in [k]} (\mathbf{x}'_{i,j} - \mathbf{x}_{i,j}) \cdot \partial_{i,j} F(\mathbf{x}) - n^2 \delta^2 M,$$

  *i.e., the difference $F(\mathbf{x}') - F(\mathbf{x})$ can be approximated by the linear (first-order Taylor) expansion at $\mathbf{x}$ with error $O(n^2 \delta^2)$.*

As a generalization of the multilinear extension of submodular functions, $k$-multilinear extension also exhibits multilinearity and non-positive Hessian elements. Furthermore, monotonicity is preserved by the extension.

Several novel properties emerge due to the inherent partition property of $k$-submodular functions. First, the Hessian of our extension contains zero-value elements in the same $i$'s blocks,

---

[3] $e_{i,j}$ is the $(i,j)$-th unit basis vector in $\mathbb{R}^{n \times k}$.

i.e. $\partial^2 F/\partial \mathbf{x}_{i_1,j_1}\partial \mathbf{x}_{i_2,j_2} = 0$ if $i_1 = i_2$, which is useful in designing rounding schemes. Our extension also exhibits an exclusive pairwise monotone property, which allows us to handle the non-monotone case. More importantly, we demonstrate a novel approximate linearity property for the $k$-submodular case, which allows us to estimate the increment of movement with a sufficient small stepsize in the analyses of the Frank-Wolfe type methods (Bian et al., 2017). This property is analogous to the approximate linearity described in Equation (4) of Bian et al. (2017) for DR-submodular maximization, which arises from the Lipschitz continuity of the gradient of DR-submodular functions. Although our bound on the elements of the Hessian implies a Lipschitz constant for the gradient of the $k$-multilinear extension, this Lipschitz constant scales with $k^2$. However, our error in the approximate linearity is independent of $k$. This independence stems from the fact that any difference $\mathbf{x} - \mathbf{x}'$ lies within $\Delta_k^n$.

*Proof of Lemma C.1.* We first remind the definition of the multilinear extension of $k$-submodular functions.

**Definition C.2** ($k$-**multilinear extension**). *Given a $k$-submodular function $f\colon \{0,\ldots,k\}^n \to \mathbb{R}_{\geq 0}$, we define its multilinear extension $F\colon \Delta_k^n \to \mathbb{R}_{\geq 0}$ as*

$$F(\mathbf{x}) = \sum_{\mathbf{s}\in\{0,\ldots,k\}^n} f(\mathbf{s}) \prod_{i\in[n]:\mathbf{s}_i\neq 0} \mathbf{x}_{i,\mathbf{s}_i} \prod_{i\in[n]:\mathbf{s}_i=0}\left(1 - \sum_{j=1}^{k}\mathbf{x}_{i,j}\right). \tag{9}$$

**Preservation of monotonicity.** As

$$\frac{\partial F}{\partial \mathbf{x}_{i,j}} = \sum_{\substack{\mathbf{s}\in\{0,\ldots,k\}^n \\ \mathbf{s}_i=j}} f(\mathbf{s}) \prod_{t\in[n]\setminus\{i\}:\mathbf{s}_t\neq 0} \mathbf{x}_{t,\mathbf{s}_t} \prod_{t\in[n]:\mathbf{s}_t=0}\left(1 - \sum_{l=1}^{k}\mathbf{x}_{t,l}\right)$$
$$- \sum_{\substack{\mathbf{s}\in\{0,\ldots,k\}^n \\ \mathbf{s}_i=0}} f(\mathbf{s}) \prod_{t\in[n]:\mathbf{s}_t\neq 0} \mathbf{x}_{t,\mathbf{s}_t} \prod_{t\in[n]\setminus\{i\}:\mathbf{s}_t=0}\left(1 - \sum_{l=1}^{k}\mathbf{x}_{t,l}\right).$$

for every vector $\mathbf{s}$ such that $\mathbf{s}_i = 0$, which is in the second term, we can find $S'$ such that
$$\mathbf{s}'_i = j \text{ and } \mathbf{s}'_l = \mathbf{s}_l \text{ for any } l \neq i,$$
in the first term. When $f$ is monotone (assume that $f$ is increasing without loss of generality), we have $f(\mathbf{s}') \geq f(\mathbf{s})$. Thus $\frac{\partial F}{\partial \mathbf{x}_{i,j}} \geq 0$, for all $i \in [n]$ and $j \in [k]$, which indicates that $F$ is also monotone as desired.

**Pairwise monotonicity.** We first remind the (discrete) $k$-submodular function holds the pairwise monotone property.

**Theorem C.3** (**Theorem 7 of Ward & Zivný (2016)**). *$k$-submodular function holds pairwise monotone property, i.e., for any $i \in [n]$, $j_1, j_2 \in [k]$, $\mathbf{s}$ with $\mathbf{s}_i = 0$, we have*
$$f(\mathbf{s} + \mathbf{e}_{j_1}) + f(\mathbf{s}_i + \mathbf{e}_{j_2}) \geq 2f(\mathbf{s}).$$

Taking derivative of Eq. (9),

$$\frac{\partial F}{\partial \mathbf{x}_{i,j_1}} + \frac{\partial F}{\partial \mathbf{x}_{i,j_2}} = \sum_{\substack{\mathbf{s}\in\{0,\ldots,k\}^n \\ \mathbf{s}_i=j_1}} f(\mathbf{s}) \prod_{t\in[n]\setminus\{i\}:\mathbf{s}_t\neq 0} \mathbf{x}_{t,\mathbf{s}_t} \prod_{t\in[n]:\mathbf{s}_t=0}\left(1 - \sum_{l=1}^{k}\mathbf{x}_{t,l}\right)$$
$$- \sum_{\substack{\mathbf{s}\in\{0,\ldots,k\}^n \\ \mathbf{s}_i=0}} f(\mathbf{s}) \prod_{t\in[n]:\mathbf{s}_t\neq 0} \mathbf{x}_{t,\mathbf{s}_t} \prod_{t\in[n]\setminus\{i\}:\mathbf{s}_t=0}\left(1 - \sum_{l=1}^{k}\mathbf{x}_{t,l}\right)$$
$$+ \sum_{\substack{\mathbf{s}\in\{0,\ldots,k\}^n \\ \mathbf{s}_i=j_2}} f(\mathbf{s}) \prod_{t\in[n]\setminus\{i\}:\mathbf{s}_t\neq 0} \mathbf{x}_{t,\mathbf{s}_t} \prod_{t\in[n]:\mathbf{s}_t=0}\left(1 - \sum_{l=1}^{k}\mathbf{x}_{t,l}\right)$$
$$- \sum_{\substack{\mathbf{s}\in\{0,\ldots,k\}^n \\ \mathbf{s}_i=0}} f(\mathbf{s}) \prod_{t\in[n]:\mathbf{s}_t\neq 0} \mathbf{x}_{t,\mathbf{s}_t} \prod_{t\in[n]\setminus\{i\}:\mathbf{s}_t=0}\left(1 - \sum_{l=1}^{k}\mathbf{x}_{t,l}\right).$$

For every vector $\mathbf{s}$ with $\mathbf{s}_i = 0$, which is in the second term and the fourth term, we can find a set tuple $\mathbf{s}^1$ such that

$$\mathbf{s}_l^1 = j_1 \text{ and } \mathbf{s}_l^1 = \mathbf{s}_l \text{ for any } l \neq i \,,$$

in the first term, and a set tuple $\mathbf{s}^2$ such that

$$\mathbf{s}_l^2 = j_2 \text{ and } \mathbf{s}_l^2 = \mathbf{s}_l \text{ for any } l \neq i \,,$$

in the third term. By the (discrete) pairwise monotonicity of $k$-submodular functions, we have

$$f(\mathbf{s}^1) - f(\mathbf{s}) + f(\mathbf{s}^2) - f(\mathbf{s}) \geq 0 \,.$$

Thus

$$\frac{\partial F}{\partial \mathbf{x}_{i,j_1}} + \frac{\partial F}{\partial \mathbf{x}_{i,j_2}} \geq 0 \,.$$

**Multilinearity.** Taking derivative of Eq. (9) with respect to $\mathbf{x}_{i,j}$,

$$\frac{\partial F}{\partial \mathbf{x}_{i,j}} = \sum_{\substack{\mathbf{s} \in \{0,\ldots,k\}^n \\ \mathbf{s}_i = j}} f(\mathbf{s}) \prod_{t \in [n] \setminus \{i\}: \mathbf{s}_t \neq 0} \mathbf{x}_{t,\mathbf{s}_t} \prod_{t \in [n]: \mathbf{s}_t = 0} \left(1 - \sum_{l=1}^k \mathbf{x}_{t,l}\right)$$

$$- \sum_{\substack{\mathbf{s} \in \{0,\ldots,k\}^n \\ \mathbf{s}_i = 0}} f(\mathbf{s}) \prod_{t \in [n]: \mathbf{s}_t \neq 0} \mathbf{x}_{t,\mathbf{s}_t} \prod_{t \in [n] \setminus \{i\}: \mathbf{s}_t = 0} \left(1 - \sum_{l=1}^k \mathbf{x}_{t,l}\right) \,.$$

As both the terms do not depend on $x_{i,j}$, the derivative is constant when other coordinates are fixed.

**Element-wise non-positive Hessian.** Taking the second-order derivative of Eq. (9) with respect to $\mathbf{x}_{i_1,j_1}$ and $\mathbf{x}_{i_2,j_2}$,

$$\frac{\partial^2 F}{\partial \mathbf{x}_{i_1,j_1} \partial \mathbf{x}_{i_2,j_2}} = \sum_{\substack{\mathbf{s} \in \{0,\ldots,k\}^n \\ \mathbf{s}_{i_1} = j_1, \, \mathbf{s}_{i_2} = j_2}} f(\mathbf{s}) \prod_{t \in [n] \setminus \{i_1,i_2\}: \mathbf{s}_t \neq 0} \mathbf{x}_{t,\mathbf{s}_t} \prod_{t \in [n]: \mathbf{s}_t = 0} \left(1 - \sum_{l=1}^k \mathbf{x}_{t,l}\right)$$

$$- \sum_{\substack{\mathbf{s} \in \{0,\ldots,k\}^n \\ \mathbf{s}_{i_1} = 0, \, \mathbf{s}_{i_2} = j_2}} f(\mathbf{s}) \prod_{t \in [n] \setminus \{i_2\}: \mathbf{s}_t \neq 0} \mathbf{x}_{t,\mathbf{s}_t} \prod_{t \in [n] \setminus \{i_1\}: \mathbf{s}_t = 0} \left(1 - \sum_{l=1}^k \mathbf{x}_{t,l}\right)$$

$$- \sum_{\substack{\mathbf{s} \in \{0,\ldots,k\}^n \\ \mathbf{s}_{i_1} = j_1, \, \mathbf{s}_{i_2} = 0}} f(\mathbf{s}) \prod_{t \in [n] \setminus \{i_1\}: \mathbf{s}_t \neq 0} \mathbf{x}_{t,\mathbf{s}_t} \prod_{t \in [n] \setminus \{i_2\}: \mathbf{s}_t = 0} \left(1 - \sum_{l=1}^k \mathbf{x}_{t,l}\right) \tag{10}$$

$$+ \sum_{\substack{\mathbf{s} \in \{0,\ldots,k\}^n \\ \mathbf{s}_{i_1} = 0, \, \mathbf{s}_{i_2} = 0}} f(\mathbf{s}) \prod_{t \in [n]: \mathbf{s}_t \neq 0} \mathbf{x}_{t,\mathbf{s}_t} \prod_{t \in [n] \setminus \{i_1,i_2\}: \mathbf{s}_t = 0} \left(1 - \sum_{l=1}^k \mathbf{x}_{t,l}\right) \,.$$

If $i_1 \neq i_2$, for every vector $\mathbf{s}$ such that $\mathbf{s}_{i_1} = 0$ and $\mathbf{s}_{i_2} = 0$ which is in the fourth term, we can find a vector $\mathbf{s}^1$ such that,

$$\mathbf{s}_{i_1}^1 = j_1 \text{ and } \mathbf{s}_i^1 = \mathbf{s}_i \text{ for any } i \neq i_1 \,,$$

in the third sum, and a vector $\mathbf{s}^2$ such that,

$$\mathbf{s}_{i_2}^2 = j_2 \text{ and } \mathbf{s}_i^2 = \mathbf{s}_i \text{ for any } i \neq i_2 \,,$$

in the second sum, and a vector $\mathbf{s}^0$ such that,

$$\mathbf{s}_{i_1}^0 = j_1, \ \mathbf{s}_{i_2}^0 = j_2 \text{ and } \mathbf{s}_i^0 = \mathbf{s}_i \text{ for any } i \notin \{i_1, i_2\} \,.$$

Thus we have $\min_0(\mathbf{s}^1, \mathbf{s}^2) = \mathbf{s}$ and $\max_0(\mathbf{s}^1, \mathbf{s}^2) = \mathbf{s}^0$. Due to $k$-submodularity, we have

$$f(\mathbf{s}^0) + f(\mathbf{s}) - f(\mathbf{s}^1) - f(\mathbf{s}^2) \leq 0 \,,$$

which implies that

$$\frac{\partial^2 F}{\partial \mathbf{x}_{i_1,j_1} \partial \mathbf{x}_{i_2,j_2}} \leq 0 \,.$$

On the other hand, due to submodularity, we have

$$|f(\mathbf{s}^0) + f(\mathbf{s}) - f(\mathbf{s}^1) - f(\mathbf{s}^2)| \le |f(\mathbf{s}^0) - f(\mathbf{s}^1)| + |f(\mathbf{s}^2) - f(\mathbf{s})| \le 2M\,,$$

which implies that

$$\frac{\partial^2 F}{\partial \mathbf{x}_{i_1,j_1} \partial \mathbf{x}_{i_2,j_2}} \ge -2M\,.$$

If $i_1 = i_2 = i$, by the multilinearity we have

$$\frac{\partial^2 F}{\partial \mathbf{x}_{i,j_1} \partial \mathbf{x}_{i,j_2}} = 0\,.$$

**Approximate linearity.** Since $F$ is polynomial in $\mathbf{x}$, by the Lagrangian form of Taylor's Theorem, $F(\mathbf{x}')$ at $F(\mathbf{x})$ can be expanded as

$$F(\mathbf{x}') - F(\mathbf{x}) = (\mathbf{x}' - \mathbf{x})^T \nabla F(\mathbf{x}) + \frac{1}{2}(\mathbf{x}' - \mathbf{x})^T H(\xi)(\mathbf{x}' - \mathbf{x}),$$

where $H(\cdot)$ is the Hessian matrix, and $\xi$ is a point that lies on the line segment connecting points $\mathbf{x}$ and $\mathbf{x}'$, Now we consider an element $\frac{\partial^2 F(\xi)}{\partial \mathbf{x}_{i_1,j_1} \partial \mathbf{x}_{i_2,j_2}}$ in $H(\xi)$. By the property of Element-wise non-positive Hessian, we have

$$\left| \frac{\partial^2 F}{\partial \mathbf{x}_{i_1,j_1} \partial \mathbf{x}_{i_2,j_2}} \right| \le 2M.$$

Therefore, if $\mathbf{x}' - \mathbf{x} \in \delta \Delta_k^n$, i.e., for all $i \in [n]$, $\sum_{j \in [k]} \mathbf{x}'_{i,j} - \mathbf{x}_{i,j} \le \delta$, we have

$$\left| \frac{1}{2}(\mathbf{x}' - \mathbf{x})^T H(\xi)(\mathbf{x}' - \mathbf{x}) \right|$$

$$\le \sum_{i_1 \in [n], j_1 \in [k]} \sum_{i_2 \in [n], j_2 \in [k]} \left| \frac{\partial^2 F}{\partial \mathbf{x}_{i_1,j_1} \partial \mathbf{x}_{i_2,j_2}} \right| |\mathbf{x}'_{i_1,j_1} - \mathbf{x}_{i_1,j_1}| |\mathbf{x}'_{i_2,j_2} - \mathbf{x}_{i_2,j_2}|$$

$$\le \frac{1}{2} \sum_{i_1 \in [n], j_1 \in [k]} \sum_{i_2 \in [n], j_2 \in [k]} 2M |\mathbf{x}'_{i_1,j_1} - \mathbf{x}_{i_1,j_1}| |\mathbf{x}'_{i_2,j_2} - \mathbf{x}_{i_2,j_2}|$$

$$= M \sum_{i_1 \in [n]} \sum_{i_2 \in [n]} \left( \sum_{j_1 \in [k]} |\mathbf{x}'_{i_1,j_1} - \mathbf{x}_{i_1,j_1}| \right) \left( \sum_{j_2 \in [k]} |\mathbf{x}'_{i_2,j_2} - \mathbf{x}_{i_2,j_2}| \right)$$

$$\le M \sum_{i_1 \in [n]} \sum_{i_2 \in [n]} \delta^2$$

$$= n^2 \delta^2 M.$$

Therefore,

$$F(\mathbf{x}') - F(\mathbf{x}) \ge \sum_{i \in [n], j \in [k]} (\mathbf{x}'_{i,j} - \mathbf{x}_{i,j}) \cdot \partial_{i,j} F(\mathbf{x}) - n^2 \delta^2 M.$$

$\square$

## D  PROOF OF LEMMA 3.2: A NOVEL ROUNDING SCHEME

In this section, we prove Lemma 3.2 by describing and analyzing a rounding algorithm called KSUBROUND (Algorithm 2).

**Useful notations and facts for Lemma 3.2.** We first recall the round procedure of submodular functions (Călinescu et al., 2011; Chekuri et al., 2014; 2010b). For $\varepsilon < 1/2$ we say $\mathcal{P}_\mathcal{K}$ is *small-weighted*, if its weight matrix $A$ satisfies $a_{i,j} < \varepsilon^3$ for all $i \in [l]$ and $j \in [n]$.

**Lemma D.1** ((Călinescu et al., 2011; Chekuri et al., 2014; 2010b)). *Assume $\mathcal{P} \subseteq [0,1]^n$ is a down-closed polytope and $G : [0,1]^n \to \mathbb{R}_{\geq 0}$ is a multilinear extension of some submodular function $g$, then there exists an algorithm, that takes a vector $\mathbf{y} \in \mathcal{P}$ and the function $G$ as input and return a set $S \in 2^n$ obeying $1_S \in \mathcal{P}$ and*

- $\mathbb{E}[G(1_S)] \geq G(\mathbf{y})$, *when $\mathcal{P}$ is single matroid constraint with rank $r$, with calling $\mathcal{O}_{\mathcal{P}}$ at most $Nr^2$ times if $\mathbf{y}$ is convex combination of $N$ bases;*

- $\mathbb{E}[G(1_S)] \geq (1-\varepsilon)G(\mathbf{y})$, *when $\mathcal{P}$ is $l = O(1)$ small-weighted knapsack constraints, with calling $\mathcal{O}_{\mathcal{P}}$ at most $\text{poly}\left(n, \frac{1}{\varepsilon}\right)$ times, for any fixed $\varepsilon > 0$;*

- $\mathbb{E}[G(1_S)] \geq \left(\frac{0.6}{b} - \varepsilon\right) G(\mathbf{y})$, *when $\mathcal{P}$ is intersection of $b$ matroid constraints and $l = O(1)$ small-weighted knapsack constraints, with calling $\mathcal{O}_{\mathcal{P}}$ at most $\text{poly}\left(n, \frac{1}{\varepsilon}\right)$ times, for any fixed $\varepsilon > 0$;*

*without calling to $g$. We refer this algorithm as* SUBROUND$(\mathbf{y}, G, \mathcal{P})$.

Our approach uses the rounding procedures SUBROUND, which are applied after reducing the multilinear extension of the $k$-submodular function to the multilinear extension of a submodular function with an index vector $\mathbf{I} \in \{1, \ldots, k\}^n$.

**Definition D.2** (**Reduced multilinear extension**). *Given a multilinear extension of $k$-submodular function, $F : \Delta_k^n \to \mathbb{R}_{\geq 0}$ for any index vector $\mathbf{I} \in \{1, \ldots, k\}^n$, we define a reduced function $F_{\mathbf{I}} : [0,1]^n \to \mathbb{R}_{\geq 0}$ as*

$$F_{\mathbf{I}}(\mathbf{x}) = F(\mathbf{x}^{\mathbf{I}}),$$

*where $\mathbf{x}^{\mathbf{I}} \in \Delta_k^n$ is defined as*

$$\mathbf{x}_{i,j}^{\mathbf{I}} = \begin{cases} \mathbf{x}_i & j = \mathbf{I}_i, \\ 0 & otherwise. \end{cases}$$

Intuitively, we define a reduced function by constraining $\mathbf{x}^{\mathbf{I}}$ to only take non-zero values at the coordinates specified by an index vector $\mathbf{I} \in \{1, \ldots, k\}^n$. Such reduced functions enjoy the submodularity shown in Claim D.3.

**Claim D.3.** *If $F$ is a multilinear extension of the $k$-submodular function $f$, the reduced function $F_{\mathbf{I}}$ is a multilinear extension of the submodular function $f_{\mathbf{I}} : 2^n \to \mathbb{R}_{\geq 0}$ defined as*

$$f_{\mathbf{I}}(S) = f(S^{\mathbf{I}}),$$

*where $S^{\mathbf{I}} \in \{0, \ldots, k\}^n$ is defined as*

$$S_i^{\mathbf{I}} = \begin{cases} \mathbf{I}_i & i \in S, \\ 0 & i \notin S. \end{cases}$$

*Proof.* We first illustrate the function-extension correspondence in the following figure.

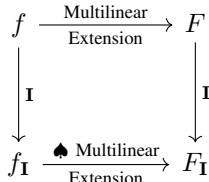

We also illustrate the domain correspondence in the below figure.

$$\begin{array}{ccc} \{0, \ldots, k\}^n & \xrightarrow{\text{Extension}} & \Delta_k^n \\ \Big\downarrow{\mathbf{I}} & & \Big\downarrow{\mathbf{I}} \\ 2^n & \xrightarrow{\text{Extension}} & [0,1]^n \end{array}$$

In the following, we complete the proof by showing the submodularity of $f_{\mathbf{I}}$ and prove the multilinear extension relationship between $f_{\mathbf{I}}$ and $F_{\mathbf{I}}$ (marked as ♠ in the first figure).

We obtain the submodularity of $f_{\mathbf{I}}$ by the inequality that

$$f_{\mathbf{I}}(S) + f_{\mathbf{I}}(T) = f(S^{\mathbf{I}}) + f(T^{\mathbf{I}}) \geq f(\min_0(S^{\mathbf{I}}, T^{\mathbf{I}})) + f(\max_0(S^{\mathbf{I}}, T^{\mathbf{I}})) = f_{\mathbf{I}}(S \cap T) + f_{\mathbf{I}}(S \cup T).$$

By the definition of the reduced function, we have

$$
\begin{aligned}
F_{\mathbf{I}}(\mathbf{x}) &= F(\mathbf{x}^{\mathbf{I}}) \\
&= \sum_{\mathbf{s} \in \{0,\dots,k\}^n} f(\mathbf{s}) \prod_{i \in [n]: \mathbf{s}_i \neq 0} \mathbf{x}^{\mathbf{I}}_{i,\mathbf{s}_i} \prod_{i \in [n]: \mathbf{s}_i = 0} \left(1 - \sum_{j=1}^k \mathbf{x}^{\mathbf{I}}_{i,j}\right) \\
&= \sum_{S \in 2^n} f(S^{\mathbf{I}}) \prod_{i \in [n]: S^{\mathbf{I}}_i \neq 0} \mathbf{x}^{\mathbf{I}}_{i,S^{\mathbf{I}}_i} \prod_{i \in [n]: S^{\mathbf{I}}_i = 0} \left(1 - \sum_{j=1}^k \mathbf{x}^{\mathbf{I}}_{i,j}\right) \\
&= \sum_{S \in 2^n} f(S^{\mathbf{I}}) \prod_{i \in [n]: S^{\mathbf{I}}_i \neq 0} \mathbf{x}_i \prod_{i \in [n]: S^{\mathbf{I}}_i = 0} \left(1 - \mathbf{x}_i\right) \\
&= \sum_{S \in 2^n} f_{\mathbf{I}}(S) \prod_{i \in S} \mathbf{x}_i \prod_{i \notin S} \left(1 - \mathbf{x}_i\right).
\end{aligned}
$$

Thus, $F_{\mathbf{I}}$ is the multilinear extension of $f_{\mathbf{I}}$.

The correspondence between the reduced function and the submodular function can also be understood through a probabilistic view. Specifically, we consider a random vector $\tilde{\mathbf{s}} \in \{0,\dots,k\}^n$, where each entry $\tilde{\mathbf{s}}_i \neq 0$ is drawn independently with probability $\sum_{j=1}^n \mathbf{x}^{\mathbf{I}}_{i,j}$ for each $i \in [n]$, and we have $\sum_{j=1}^n \mathbf{x}^{\mathbf{I}}_{i,j} = \mathbf{x}^{\mathbf{I}}_{i,\mathbf{I}_i} = \mathbf{x}_i$. If $\tilde{\mathbf{s}}_i \neq 0$ in this process, $\mathbf{s}_i$ is assigned with value $\mathbf{I}_i$. We can observe that this probability-based definition of reduced function is equivalent to the probability-based definition of the multilinear extension.

□

Now we are ready to use the rounding scheme for submodular maximization (Lemma D.1) to design a rounding scheme for $k$-submodular maximization (Lemma D.4).

**Lemma D.4.** *Given a non-monotone $k$-submodular function $f$, its multilinear extension $F$, a support constraint $\mathcal{P}$, a fractional solution $\mathbf{x} \in \Delta_k^n$ with $\mathbf{x} \sim \mathcal{P}$, there is an algorithm* KSUBROUND$(\mathbf{x}, F, \mathcal{P})$ *which runs in polynomial time and outputs an integral solution $\mathbf{s} \in \{0,\dots,k\}^n$ with $\mathbf{s} \sim \mathcal{P}$ such that*

- $\mathbb{E}[f(\mathbf{s})] \geq F(\mathbf{x})$, *when $\mathcal{P}$ is single matroid constraint with rank $r$, with calling $\mathcal{O}_{\mathcal{P}}$ at most $Nr^2$ times if $\mathbf{x}^{\mathbf{I}}$ is convex combination of $N$ bases;*

- $\mathbb{E}[f(\mathbf{s})] \geq (1-\varepsilon)F(\mathbf{x})$ *when $\mathcal{P}$ is $l = O(1)$ small-weighted knapsack constraints, with calling $\mathcal{O}_{\mathcal{P}}$ at most $\mathrm{poly}\left(n, \frac{1}{\varepsilon}\right)$ times, for any fixed $\varepsilon > 0$;*

- $\mathbb{E}[f(\mathbf{s})] \geq \left(\frac{0.6}{b}F(\mathbf{x}) - \varepsilon\right)$ *when $\mathcal{P}$ is intersection of $b$ matroid constraints and $l = O(1)$ small-weighted knapsack constraints, with calling $\mathcal{O}_{\mathcal{P}}$ at most $\mathrm{poly}\left(n, \frac{1}{\varepsilon}\right)$ times, for any fixed $\varepsilon > 0$.*

**Algorithm for Lemma D.4.** Now we are ready to introduce our rounding algorithm KSUBROUND (Algorithm 2) which consists of three phases: rounding from $\Delta_k^n$ to $[0,1]^n$ (Lines 1-7), rounding from $[0,1]^n$ to $2^n$ (Line 8) and recovering from $2^n$ to $(k+1)^n$ (Lines 9-12). In each phase, we preserve the feasibility and control the loss.

In the first phase (Line 1-7), for any $i$, we merge all non-zero values and assign value $j$ to the $i$-th coordinate of the index vector $\mathbf{I}$ with a categorical probability of the proportion. This merging process does not conflict with the support constraint because the sum $\sum_{j=1}^k \mathbf{x}_{i,j}$ remains constant.

Furthermore, at each iteration $i$, by the definition of multilinear extension $F$, the function value is exactly the linear combination of function value at every vertex of the affined corner of cube, i.e., [4]

$$
\begin{aligned}
\mathbb{E}_{\mathbf{I}}\left[F_{\mathbf{I}}(\mathbf{y})\right] &= \mathbb{E}_{\mathbf{I}}\left[F(\mathbf{y}^{\mathbf{I}})\right] \\
&= \mathbb{E}_{\mathbf{I}}\left[\sum_{\mathbf{s}\in\{0,\ldots,k\}^n} f(\mathbf{s}) \prod_{i\in[n]:\mathbf{s}_i\neq 0} \mathbf{y}^{\mathbf{I}}_{i,\mathbf{s}_i} \prod_{i\in[n]:\mathbf{s}_i=0}\left(1-\sum_{j=1}^k \mathbf{y}^{\mathbf{I}}_{i,j}\right)\right] \\
&= \sum_{\mathbf{s}\in\{0,\ldots,k\}^n} f(\mathbf{s}) \prod_{i\in[n]:\mathbf{s}_i\neq 0} \mathbf{x}_{i,\mathbf{s}_i} \prod_{i\in[n]:\mathbf{s}_i=0}\left(1-\sum_{j=1}^k \mathbf{x}_{i,j}\right) \\
&= F(\mathbf{x}).
\end{aligned}
$$

In the second phase (Line 14), we apply the rounding procedure SUBROUND which takes the fractional solution $\mathbf{y} \in \mathcal{P} \subseteq [0,1]^n$ and the reduced function $F_{\mathbf{I}}$ as input and returns an integer solution $S \in 2^n$. The loss of this rounding procedure is bounded by Lemma D.1.

In the final phase (Lines 16-18), we recover the solution $\mathbf{s} \in \Delta_k^n$ by setting $\mathbf{s}_i = \mathbf{I}_i$ if $i \in S$. This recovery step incurs no loss since the recovery procedure and reducing procedure correspond to the same index vector $\mathbf{I}$.

---

**Algorithm 2:** $\text{KSUBROUND}(\mathbf{x}, F, \mathcal{P})$

---

**Input :** A fractional solution $\mathbf{x} \in \Delta_k^n$ with $\mathbf{x} \sim \mathcal{P}$, $\mathcal{O}_{F,\nabla F}$, membership oracle $\mathcal{O}_{\mathcal{P}}$.

1 **Initialize** $\mathbf{y} \leftarrow [0,\ldots,0]^\top \in [0,1]^n$ and $\mathbf{I} \leftarrow [0,\ldots,0]^\top \in \{0,\ldots,k\}^n$.

2 **for** $i \in [n]$ **do**

3      $\mathbf{y}_i \leftarrow \sum_{j=1}^k \mathbf{x}_{i,j}$.

4      **if** $\mathbf{y}_i \neq 0$ **then**

5          **With Categorical Probability** $p = \mathbf{x}_{i,j}/\sum_{j=1}^k \mathbf{x}_{i,j}$: $\mathbf{I}_i \leftarrow j$.

6      **else**

7          $\mathbf{I}_i \leftarrow 0$.

8 $S \leftarrow \text{SUBROUND}(\mathbf{y}, F_{\mathbf{I}}, \mathcal{P})$.

9 Initialize $\mathbf{s} \leftarrow [0,\ldots,0]^\top \in \{0,\ldots,k\}^n$.

10 **for** $i \in [n]$ **do**

11      **if** $i \in S$ **then**

12          $\mathbf{s}_i \leftarrow \mathbf{I}_i$.

13 **Return:** $\mathbf{s}$.

---

*Proof of Lemma D.4.* We first analyze the feasibility and then prove the approximation performance.

**Feasibility** In Lines 1-7, since $\mathbf{x} \sim \mathcal{P} = \left\{ \mathbf{x} \in \Delta_k^n : \left(\sum_{j=1}^k \mathbf{x}_{1,j}, \ldots, \sum_{j=1}^k \mathbf{x}_{n,j}\right)^\top \in \mathcal{P}\right\}$, and the

sum $\sum_{j=1}^k \mathbf{x}_{i,j}$ will remains constant during the moving for any $i$, we have $\mathbf{y} \sim \mathcal{P}$. In Line 8, by Lemma D.1, we have $1_S \in \mathcal{P}$. In Line 9-12, by the definition of support constraint, we have $\mathbf{s} \sim \mathcal{P}$.

**Approximation ratio of Algorithm 2.** In Line 1-7, by the definition of $F$ and $F^{\mathbf{I}}$, we conclude that

$$\mathbb{E}_{\mathbf{I}}\left[F^{\mathbf{I}}(\mathbf{y})\right] = F(\mathbf{x}). \tag{11}$$

---

[4] We can also conclude the linearity by the zero value of the Hessian element at the same element $i$'s block, i.e., $\frac{\partial^2 F}{\partial x_{i,j_1} \partial x_{i,j_2}} = 0$ for any $j_1, j_2 \in [k]$.

In Lines 9-12, by the definition of $F_{\mathbf{I}}$, we have

$$F(1_{\mathbf{s}}) = F_{\mathbf{I}}(1_S). \tag{12}$$

Combine Eq. equation 11 and 12 and Lemma D.1, we complete the proof. $\square$

Finally, we combine Lemma D.4 and standard enumeration tricks to prove the Lemma 3.2.

*Proof of Lemma 3.2.* For the matroid constraint, we directly employ $\mathtt{KSUBROUND}(\mathbf{x}, F, \mathcal{P})$ on the vector $\mathbf{x}$, which is the output from $\mathtt{FW}(f, \mathcal{P}, \varepsilon, \delta)$. We note the reduced vector $\mathbf{x}^{\mathbf{I}}$ of the output $\mathbf{x}$ from $\mathtt{FW}(f, \mathcal{P}, \varepsilon, \delta)$ is convex combination of $N$ bases.

In the case involving $l = O(1)$ knapsack constraints (and possibly other constraints), we implement an enumeration strategy for every subset $A \subseteq [n]$ with $|A| < n_0$ where $n_0 = \frac{2}{\varepsilon^4}$. Let $\bar{A} = n \backslash A$. For each $A$, we explore all potential vectors in $\{0, \ldots, k\}^A$. For any $\mathbf{s}_A \in \{0, \ldots, k\}^A$ and $\mathbf{s}_{\bar{A}} \in \{0, \ldots, k\}^{\bar{A}}$, we define the concatenate vector as $\oplus(\mathbf{s}_A, \mathbf{s}_{\bar{A}})$ such that $\oplus(\mathbf{s}_A, \mathbf{s}_{\bar{A}})_i = \begin{cases} (\mathbf{s}_A)_i & \text{if } i \in A, \\ (\mathbf{s}_{\bar{A}})_i & \text{if } i \in \bar{A}. \end{cases}$ For each vector $\mathbf{s}_A \in \{0, \ldots, k\}^A$, we define the resident $k$-submodular maximization problem such that the objective function $f_{\mathbf{s}_A} : \{0, \ldots, k\}^{\bar{A}} \to \mathbb{R}_{\geq 0}$ satisfies that $f_{\mathbf{s}_A}(\mathbf{s}') = f(\oplus(\mathbf{s}_A, \mathbf{s}'))$. By definition, the objective function $f_{\mathbf{s}_A}$ still satisfies the $k$-submodularity. For the constraint, the vector $\mathbf{s}'$ is deemed feasible iff $\oplus(\mathbf{s}_A, \mathbf{s}')$ is feasible for $\mathcal{P}$, and refer its conjunction constraint as $\mathcal{P}_{\mathbf{s}_A}$. For each resident $k$-submodular maximization problem, we perform $\mathtt{FW}(f_{\mathbf{s}_A}, \mathcal{P}_{\mathbf{s}_A}, \varepsilon, \delta)$ to solve the continuous problem and yields a fractional solution $\mathbf{x}(\mathbf{s}_A)$. Then we perform rounding scheme $\mathtt{KSUBROUND}(\mathbf{x}(\mathbf{s}_A), F_{\mathbf{s}_A}, \mathcal{P}_{\mathbf{s}_A})$ to obtain a solution $\mathbf{s}_{\bar{A}}(\mathbf{s}_A) \in \{0, \ldots, k\}^{\bar{A}}$; note that if the knapsack constraint is not small-weighted, we can still perform a rounding scheme $\mathtt{KSUBROUND}$ but with a weaker guarantee or without a guarantee. Finally, we output the best solution overall cases by comparing all $f(\oplus(\mathbf{s}_A, \mathbf{s}_{\bar{A}}(\mathbf{s}_A)))$.

Now we consider a special case of $\mathbf{s}_A$. Select $\mathbf{s}_A$ greedily from the optimal solution, by picking elements as long as their marginal contribution is at least $\varepsilon^4 OPT$; note that $|A| \leq \frac{1}{\varepsilon^4}$. For any $i \in \bar{A}$, we add it randomly to $\mathbf{s}_A$ if its size for some knapsack constraint is more than $\frac{1}{\varepsilon^3}$, i.e. $a_{i,j} \geq \frac{1}{\varepsilon^3}$ for some $j \in [l]$. The number of such elements in a knapsack can be at most $\varepsilon^3$ and hence they can contribute at most $\varepsilon OPT$, and the total lost value is at most $l\varepsilon OPT$. For this $\mathbf{s}_A$, we obtain a $(\alpha(1 - (l+1)\varepsilon))$-approximate solution, when $\mathcal{P}$ is $O(1)$ knapsack constraints and a $\left(\frac{0.6\alpha}{b}(1 - (l+1)\varepsilon)\right)$-approximate solution, when $\mathcal{P}$ is the intersection of $b$ matroid constraints and $O(1)$ knapsack constraints. By rescale $\varepsilon$ we obtain the result. $\square$

## E  PROOF OF THEOREM 3.1: PERFORMANCE ANALYSIS OF ALGORITHM 1

It suffices to prove the following key lemma. By the selection of $\delta$ in Algorithm 1, Theorem 3.1 is a direct corollary of Lemmas 3.2 and 3.3.

**Lemma E.1** (**Restatement of Lemma 3.3**). *Let* $\mathbf{o}^\star = \arg\max_{\mathbf{x} \in \Delta_k^n, \mathbf{x} \sim \mathcal{P}} F(\mathbf{x})$. *If $f$ is monotone, then* $F(\mathbf{x}(1)) \geq \left(\frac{1}{2} - 2\varepsilon\right) F(\mathbf{o}^\star)$, *with probability at least $1 - \eta$.*

The key idea of Lemma 3.3 is to analyze the value gain of each iteration. Following the commonly used idea to $k$-submodular maximization (Iwata et al., 2016; Ohsaka & Yoshida, 2015; Sakaue, 2017), we construct an auxiliary sequence $\mathbf{o}(t) = \mathbf{x}(t) + (1-t)\mathbf{o}^\star$ to be a linear combination of $\mathbf{o}^\star$ and $\mathbf{x}(t)$ such that $\mathbf{o}(t)$ is still consistent to $\mathcal{P}$. Such sequence satisfies that $\mathbf{o}(0) = \mathbf{o}^\star$ and $\mathbf{o}(1) = \mathbf{x}(1)$. Then it suffices to show the decrease of the auxiliary sequence $F(\mathbf{o}(t)) - F(\mathbf{o}(t+\delta))$ is smaller than the increase of the solution sequence $F(\mathbf{x}(t+\delta)) - F(\mathbf{x}(t))$ with a additional error bounded by $O(\varepsilon M \delta)$.

*Proof of Lemma E.1.* To obtain the guarantee, we construct the following auxiliary sequences. Let

$$\mathbf{o}(t) = \mathbf{x}(t) + (1-t)\,\mathbf{o}^\star,$$

$$\mathbf{o}(t+\delta) = \mathbf{x}(t) + \delta\mathbf{v}(t) + (1-t-\delta)\,\mathbf{o}^\star,$$

and
$$\mathbf{o}'(t) = \mathbf{x}(t) + (1 - t - \delta)\,\mathbf{o}^{\star}.$$

By definition, it is clear that $\mathbf{o}(0) = \mathbf{o}^{\star}$ and $\mathbf{o}(1) = \mathbf{x}(1)$. By induction on $t$ and the definition of $\mathbf{x}(t)$, we obtain $\frac{1}{t}\mathbf{x}(t) = \sum_{i=1}^{t/\delta} \frac{\delta}{t}\mathbf{v}(i)$. Thus, $\frac{1}{t}\mathbf{x}(t)$ can be expressed as a linear combination of $\mathbf{v}(1), \ldots, \mathbf{v}(t)$. Since $\mathbf{v}(t) \sim \mathcal{P}$, it follows that $\mathbf{x}(t) \sim t \cdot \mathcal{P}$, which implies that $\mathbf{o}(t), \mathbf{o}(t+\delta) \sim \mathcal{P}$. By the definition of $\mathbf{o}'(t)$ and $\mathbf{o}(t+\delta)$, we have

$$\mathbf{o}(t+\delta) - \mathbf{o}'(t) = \delta\mathbf{v}(t) \in [0,1]^{nk}.$$

Combining the monotonicity of $F$, we have

$$F(\mathbf{o}'(t)) - F(\mathbf{o}(t+\delta)) \leq 0. \tag{13}$$

We also bound the error caused by the stochastic estimate of gradient $\widehat{\nabla F(\mathbf{x})}$.

**Lemma E.2.** *For any direction $\mathbf{y} \in \Delta_k^n$ with $\mathbf{y} \sim \mathcal{P}$ and any $t$, with probability at least $1 - \frac{\varepsilon\eta}{n^2+1}$,*

$$\langle \nabla F(\mathbf{x}(t)), \mathbf{y} \rangle \leq \langle \nabla F(\mathbf{x}(t)), \mathbf{v}(t) \rangle + 2\varepsilon M.$$

*Proof.* For any $t$

$$
\begin{aligned}
\langle \nabla F(\mathbf{x}(t)), \mathbf{y} \rangle &= \langle \widehat{\nabla F(\mathbf{x}(t))}, \mathbf{y} \rangle + \langle \nabla F(\mathbf{x}(t)) - \widehat{\nabla F(\mathbf{x}(t))}, \mathbf{y} \rangle \\
&\leq \langle \widehat{\nabla F(\mathbf{x}(t))}, \mathbf{v}(t) \rangle + \langle \nabla F(\mathbf{x}(t)) - \widehat{\nabla F(\mathbf{x}(t))}, \mathbf{y} \rangle && \text{(by choice of } \mathbf{v}(t)) \\
&= \langle \nabla F(\mathbf{x}(t)), \mathbf{v}(t) \rangle + \langle \nabla F(\mathbf{x}(t)) - \widehat{\nabla F(\mathbf{x}(t))}, \mathbf{y} \rangle + \langle \widehat{\nabla F(\mathbf{x}(t))} - \nabla F(\mathbf{x}(t)), \mathbf{v}(t) \rangle \\
&\leq \langle \nabla F(\mathbf{x}(t)), \mathbf{v}(t) \rangle + \left\| \nabla F(\mathbf{x}(t)) - \widehat{\nabla F(\mathbf{x}(t))} \right\|_2 \|\mathbf{y}\|_2 \\
&\quad + \left\| \nabla F(\mathbf{x}(t)) - \widehat{\nabla F(\mathbf{x}(t))} \right\|_2 \|\mathbf{v}(t)\|_2 && \text{(by Cauchy–Schwarz inequality)} \\
&\leq \langle \nabla F(\mathbf{x}(t)), \mathbf{v}(t) \rangle + \frac{\varepsilon M}{\sqrt{n}} \|\mathbf{y}\|_2 + \frac{\varepsilon M}{\sqrt{n}} \|\mathbf{v}(t)\|_2 && \text{(by Lemma 2.5)} \\
&\leq \langle \nabla F(\mathbf{x}(t)), \mathbf{v}(t) \rangle + 2\varepsilon M.
\end{aligned}
$$

$\square$

Now we bound the improvement in every step.

$$
\begin{aligned}
&F(\mathbf{o}(t)) - F(\mathbf{o}(t+\delta)) \\
&= F(\mathbf{o}(t)) - F(\mathbf{o}'(t)) + F(\mathbf{o}'(t)) - F(\mathbf{o}(t+\delta)) \\
&\leq F(\mathbf{o}(t)) - F(\mathbf{o}'(t)) && \text{(by Eq. (13))} \\
&= \langle \nabla F(\mathbf{o}'(t)), \mathbf{o}^{\star} \rangle \delta + \frac{1}{2}(\mathbf{o}^{\star})'H(\xi)\mathbf{o}^{\star}\delta^2 && \text{(with } \xi = \mathbf{o}'(t) + c\mathbf{o}^{\star}\delta \text{ and } c \in (0,1)) \\
&\leq \langle \nabla F(\mathbf{o}'(t)), \mathbf{o}^{\star} \rangle \delta && \text{(by element-wise non-positive Hessian)} \\
&= \sum_{i,j} \partial_{i,j} F(\mathbf{o}'(t))\mathbf{o}^{\star}_{i,j}\delta \\
&\leq \sum_{i,j} \partial_{i,j} F(\mathbf{x}(t))\mathbf{o}^{\star}_{i,j}\delta && \text{(by element-wise non-positive Hessian)} \\
&= \langle \nabla F(\mathbf{x}(t)), \mathbf{o}^{\star} \rangle \delta \\
&\leq \langle \nabla F(\mathbf{x}(t)), \mathbf{v}(t) \rangle \delta + 2\varepsilon M\delta && \text{(by Lemma E.2)} \\
&\leq F(\mathbf{x}(t+\delta)) - F(\mathbf{x}(t)) + n^2\delta^2 M + 2\varepsilon M\delta && \text{(by approximate linearity)} \\
&\leq F(\mathbf{x}(t+\delta)) - F(\mathbf{x}(t)) + 3\varepsilon M\delta. && \text{(by choice of } \delta)
\end{aligned}
$$

By Lemma E.2, the above inequality holds with probability at least $1 - \frac{\varepsilon\eta}{n^2+1}$. Thus, by union bound over $N = \lceil \frac{n^2}{\varepsilon} \rceil$ steps, we conclude that the following inequality holds

$$F(\mathbf{o}(0)) - F(\mathbf{o}(1)) \leq F(\mathbf{x}(1)) - F(\mathbf{x}(0)) + 3\varepsilon M$$

with probability at least $1 - \eta$. Thus

$$F(\mathbf{x}(1)) \geq \frac{1}{2}F(\mathbf{o}^\star) - 2\varepsilon M \geq \left(\frac{1}{2} - 2\varepsilon\right)F(\mathbf{o}^\star).$$

$\square$

Now we prove Theorem 1 by combining with Lemma 3.2.

*Proof.* Combining Lemma 3.3 and Lemma 3.2, we can show the approximation ratio part of Theorem 3.1.

Now, we analyze the query complexity and success probability of Algorithm 1. Algorithm 1 queries $\mathcal{O}_{\nabla F}$ a total of $N = \lceil \frac{n^2}{\varepsilon} \rceil$ times, which implies that the query complexity with respect to $f$ is bounded as

$$\#\text{Calls to } \mathcal{O}_f \leq \lceil \frac{n^2}{\varepsilon} \rceil \cdot \lceil \frac{16kn^4 \log\left(\frac{n^2+1}{\varepsilon\eta}\right)}{\varepsilon^2} \rceil = O\left(\frac{kn^6 \log(\frac{n}{\varepsilon\eta})}{\varepsilon^3}\right).$$

Also the query complexity of $\mathcal{O}_{\mathcal{P}}$ is bounded as

$$\#\text{Calls to } \mathcal{O}_{\mathcal{P}} \leq O\left(k^2 n^2\right) \cdot \lceil \frac{16kn^4 \log\left(\frac{n^2+1}{\varepsilon\eta}\right)}{\varepsilon^2} \rceil = O\left(\frac{k^3 n^6 \log(\frac{n}{\varepsilon\eta})}{\varepsilon^2}\right).$$

To obtain $1 - \eta$ success probability, we scale the success probability of the Frank-Wolfe algorithm FW, and the total query complexity is at most scaled by a logarithmic factor by Lemma 2.5. $\square$

## F  RESULTS FOR NON-MONOTONE $k$-SUBMODULAR MAXIMIZATION

In this section, we present an algorithm (Algorithm 3) and its analysis (Theorem F.1) for the non-monotone $k$-submodular objective. Recall that $M = \max\{\max_{i,j} F(\mathbf{e}_{i,j}) - F(\mathbf{0}), 0\}$.

**Theorem F.1** (**Main theorem II, non-monotone case**). *There exists a polynomial-time algorithm that given a non-monotone $k$-submodular $f : \{0, 1, \ldots, k\}^n$ and a constraint polytope $\mathcal{P} \subseteq \Delta_k^n$, with probability at least $1 - \eta$, outputs a solution that is*

- *$(\frac{1}{3} - \varepsilon)$-approximate under a single matroid constraint, with calling $\mathcal{O}_f$ at most $O\left(\frac{kn^6 \log(\frac{n}{\varepsilon\eta})}{\varepsilon^3}\right)$ times and calling $\mathcal{O}_{\mathcal{P}}$ at most $O\left(\frac{k^3 n^6 \log(\frac{n}{\varepsilon\eta})}{\varepsilon^2}\right)$, for any fixed $\varepsilon > 0$;*

- *$(\frac{1}{3} - \varepsilon)$-approximate under the intersection of $O(1)$ knapsack constraints, with calling $\mathcal{O}_f$ at most $O\left(k^{\text{poly}(\frac{1}{\varepsilon})} n^{\text{poly}(\frac{1}{\varepsilon})}\right)$ times and calling $\mathcal{O}_{\mathcal{P}}$ at most $O\left(k^{\text{poly}(\frac{1}{\varepsilon})} n^{\text{poly}(\frac{1}{\varepsilon})}\right)$, for any fixed $\varepsilon > 0$;*

- *$(\frac{0.2}{b} - \varepsilon)$-approximate under the intersection of $b$ matroid constraints and $O(1)$ knapsack constraints, with calling $\mathcal{O}_f$ at most $O\left(k^{\text{poly}(\frac{1}{\varepsilon})} n^{\text{poly}(\frac{1}{\varepsilon})}\right)$ times and calling $\mathcal{O}_{\mathcal{P}}$ at most $O\left(k^{\text{poly}(\frac{1}{\varepsilon})} n^{\text{poly}(\frac{1}{\varepsilon})}\right)$, for any fixed $\varepsilon > 0$.*

---

**Algorithm 3:** Frank-Wolfe algorithm for non-monotone case, $\text{NFW}(f, \mathcal{P}, \varepsilon, \eta)$

---

**Input :** Parameters $\varepsilon, \eta \in (0,1)$; oracles $\mathcal{O}_f, \mathcal{O}_{\nabla F}^{(\varepsilon, \eta)}, \mathcal{O}_{\mathcal{P}}$.

1 **Initialize:** $\mathbf{x}(0) \leftarrow \mathbf{0}$, $t \leftarrow 0$; stepsize $\delta = \frac{1}{N}$ with $N = \lceil \frac{n^2}{\varepsilon} \rceil$.

2 **while** $t < 1$ **do**

3     Find a direction $\mathbf{v}(t) = \arg\max_{\mathbf{v} \in \Delta_k^n, \mathbf{v} \sim \mathcal{P}} \langle \widehat{\nabla F(\mathbf{x}(t))}, \mathbf{v} \rangle$.          ▷ By LP

4     **Initialize:** $\mathbf{v}'(t) = \mathbf{0}$.

5     **for** $i \in supp(\mathbf{v}(t))$ **do**

6        Order partial derivative as $\partial_{i, j_1} \widehat{F(\mathbf{x}(t))} \geq \partial_{i, j_2} \widehat{F(\mathbf{x}(t))} \geq \ldots \geq \partial_{i, j_k} \widehat{F(\mathbf{x}(t))}$.

7        **if** $\partial_{i, j_2} \widehat{F(\mathbf{x}(t))} \geq 0$ **then**

8           $\mathbf{v}'_{i, j_2}(t) \leftarrow \sum_{j \in [k]} \mathbf{v}_{i,j}(t)$.

9     $\widehat{\mathbf{v}}(t) = \frac{1}{2} (\mathbf{v}(t) + \mathbf{v}'(t))$.

10     $\mathbf{x}(t + \delta) = \mathbf{x}(t) + \delta \widehat{\mathbf{v}}(t)$, $t \leftarrow t + \delta$.

11 **return s**

---

Similar to Algorithm 1, Algorithm 3 is also A Frank-Wolfe-type method that computes a fraction solution $\mathbf{x}(1) \sim \mathcal{P}$. The main difference is in the first stage, where Algorithm 3 moves along the complemented direction $\widehat{\mathbf{v}}(t)$ as an average of the locally optimal direction $\mathbf{v}(t)$ and vector $v'(t)$ depending on the signal of the second largest partial derivatives $\partial_{i, j_2} F(\mathbf{x}(t))$ for every $i \in [n]$. This construction is motivated by the pairwise monotonicity of $F$, which enables us to reduce the non-monotone case to the monotone one in the analysis. Now we prove Theorem F.1. Similar to Lemma 3.3, we first summarize the quality of the fractional solution $\mathbf{x}(1)$ in the following lemma.

**Lemma F.2.** *Let* $\mathbf{o}^\star = \arg\max_{\mathbf{x} \in \Delta_k^n, \mathbf{x} \sim \mathcal{P}} F(\mathbf{x})$. *Then* $F(\mathbf{x}(1)) \geq \left( \frac{1}{3} - 2\varepsilon \right) F(\mathbf{o}^\star)$, *with probability at least* $1 - \eta$.

*Proof.* To obtain the guarantee, we construct the following auxiliary sequences. Let $\mathbf{o}(t) = \mathbf{x}(t) + (1-t)\mathbf{o}^\star$, $\mathbf{o}(t+\delta) = \mathbf{x}(t) + \delta\widehat{\mathbf{v}}(t) + (1-t-\delta)\mathbf{o}^\star$, and $\mathbf{o}'(t) = \mathbf{x}(t) + (1-t-\delta)\mathbf{o}^\star$. Now we consider a fixed $i \in [n]$ in Line 6. Note that the feasibility of support constraint $\mathcal{P}$ is only affected by $\sum_{j \in [k]} \mathbf{v}_{i,j}$. Hence, we have $\mathbf{v}_{i,j}(t) = \begin{cases} \sum_{j' \in [k]} \mathbf{v}_{i,j'}(t) & j = j_1, \\ 0 & j \neq j_1, \end{cases}$ by the definition of $j_1$.[5] Next, we discuss two cases based on the signal of $\partial_{i, j_2} F(\mathbf{x}(t))$ at each step. We remind the concentration property of the gradient estimators: for all $i \in [n]$ and $j \in [k]$ (Lemma 2.5),

$$\left| \widehat{\partial_{i,j} F(\mathbf{x})} - \partial_{i,j} F(\mathbf{x}) \right| \leq \frac{\varepsilon M}{n\sqrt{k}} \quad \text{and} \quad \left\| \widehat{\nabla F(\mathbf{x})} - \nabla F(\mathbf{x}) \right\|_2 \leq \frac{\varepsilon M}{\sqrt{n}} \tag{14}$$

with probability at least $1 - \frac{\varepsilon\eta}{n^2+1}$.

**Case 1:** $\widehat{\partial_{i, j_2} F(\mathbf{x}(t))} < 0$. By submodularity, we have $\partial_{i, j_2} F(\mathbf{o}'(t)) \leq \partial_{i, j_2} F(\mathbf{x}(t)) < 0$. Combining Eq. (14), we have

$$\widehat{\partial_{i, j_2} F(\mathbf{o}'(t))} \leq \partial_{i, j_2} F(\mathbf{o}'(t)) + \frac{\varepsilon M}{n\sqrt{k}} \leq \widehat{\partial_{i, j_2} F(\mathbf{x}(t))} + \frac{2\varepsilon M}{n\sqrt{k}} < \frac{2\varepsilon M}{n\sqrt{k}}.$$

By pairwise monotonicity (Lemma C.1), we have

$$\partial_{i, j_1} F(\mathbf{o}'(t)) + \partial_{i, j_2} F(\mathbf{o}'(t)) \geq 0.$$

Combining Eq. (14), we have

$$\widehat{\partial_{i, j_1} F(\mathbf{o}'(t))} + \widehat{\partial_{i, j_2} F(\mathbf{o}'(t))} \geq -\frac{2\varepsilon M}{n\sqrt{k}}.$$

Thus $\partial_{i, j_1} F(\mathbf{o}'(t)) \geq -\frac{4\varepsilon M}{n\sqrt{k}}$. Furthermore, we have $\langle \nabla_i F(\mathbf{o}'(t)), \widehat{\mathbf{v}}_i(t) \rangle \geq -\frac{4\varepsilon M}{n\sqrt{k}}$. Moreover, due to the fact that $\widehat{\mathbf{v}}_i(t) = \frac{1}{2} \mathbf{v}_i(t)$, we have

$$\langle \nabla_i F(\mathbf{x}(t)), \mathbf{v}_i(t) \rangle = 2\langle \nabla_i F(\mathbf{x}(t)), \widehat{\mathbf{v}}_i(t) \rangle.$$

---

[5] Given the equivalency of the support constraint for all $\mathbf{v}_{i,j}(t)$, there should only be one unique non-zero value $j_1$, ensuring the auxiliary linear function achieves its maximum.

**Case 2:** $\partial_{i,j_2}\widehat{F(\mathbf{x}(t))} \geq 0$. By the definition of $\widehat{\mathbf{v}}(t)$ and $\mathbf{v}'(t)$, we have

$$2\widehat{\mathbf{v}}_{i,j}(t) = \mathbf{v}_{i,j}(t) + \mathbf{v}'_{i,j}(t) = \begin{cases} \sum_{j'\in[k]} \mathbf{v}_{i,j'}(t) & j = j_1, \\ \sum_{j'\in[k]} \mathbf{v}_{i,j'}(t) & j = j_2, \\ 0 & \text{otherwise.} \end{cases}$$

Combining the pairwise monotonicity that $\partial_{i,j_1}F(\mathbf{o}'(t)) + \partial_{i,j_2}F(\mathbf{o}'(t)) \geq 0$, we have

$$\langle \nabla_i F(\mathbf{o}'(t)), \widehat{\mathbf{v}}_i(t) \rangle \geq 0.$$

By the condition that $\partial_{i,j_2}\widehat{F(\mathbf{x}(t))} \geq 0$ and Eq. (14), we have

$$\partial_{i,j_2}F(\mathbf{x}(t)) \geq -\frac{\varepsilon M}{n\sqrt{k}},$$

which implies that

$$\langle \nabla_i\widehat{F(\mathbf{x}(t))}, \mathbf{v}'_i(t) \rangle \geq -\frac{\varepsilon M}{n\sqrt{k}}.$$

Combining the fact that $\widehat{\mathbf{v}}_i(t) = \frac{1}{2}\mathbf{v}_i(t) + \frac{1}{2}\mathbf{v}'_i(t)$, we have

$$\langle \nabla_i F(\mathbf{x}(t)), \mathbf{v}_i(t) \rangle \leq \langle \nabla_i F(\mathbf{x}(t)), \mathbf{v}_i(t) \rangle + \langle \nabla_i F(\mathbf{x}(t)), \mathbf{v}'_i(t) \rangle + \frac{\varepsilon M}{n\sqrt{k}}$$

$$= 2\langle \nabla_i F(\mathbf{x}(t)), \widehat{\mathbf{v}}_i(t) \rangle + \frac{\varepsilon M}{n\sqrt{k}}.$$

Combining these two cases and the approximate linearity, we have

$$F(\mathbf{o}'(t)) - F(\mathbf{o}(t+\delta)) \leq -\langle \nabla F(\mathbf{o}'(t)), \widehat{\mathbf{v}}(t) \rangle\delta + n^2\delta^2 M \leq \frac{4\varepsilon M}{\sqrt{k}} + n^2\delta^2 M, \qquad (15)$$

and

$$\langle \nabla F(\mathbf{x}(t)), \mathbf{v}(t) \rangle \leq 2\langle \nabla F(\mathbf{x}(t)), \widehat{\mathbf{v}}(t) \rangle + \frac{\varepsilon M}{\sqrt{k}}. \qquad (16)$$

By definition $\mathbf{o}(1) = \mathbf{x}(1)$ and $\mathbf{o}(0) = \mathbf{o}^\star$. We bound the improvement in every step by the following inequalities.

$$F(\mathbf{o}(t)) - F(\mathbf{o}(t+\delta))$$
$$= F(\mathbf{o}(t)) - F(\mathbf{o}'(t)) + F(\mathbf{o}'(t)) - F(\mathbf{o}(t+\delta))$$
$$\leq F(\mathbf{o}(t)) - F(\mathbf{o}'(t)) + \frac{4\varepsilon\delta}{\sqrt{k}} + n^2\delta^2 M \qquad \text{(by Eq. (15))}$$
$$= \langle \nabla F(\mathbf{o}'(t)), \mathbf{o}^\star \rangle\delta + \frac{1}{2}(\mathbf{o}^\star)'H(\xi)\mathbf{o}^\star\delta^2 + \frac{4\varepsilon\delta}{\sqrt{k}} + n^2\delta^2 M$$
$$\leq \langle \nabla F(\mathbf{o}'(t)), \mathbf{o}^\star \rangle\delta + \frac{4\varepsilon M\delta}{\sqrt{k}} + n^2\delta^2 M \qquad \text{(by element-wise non-positive Hessian)}$$
$$\leq \langle \nabla F(\mathbf{x}(t)), \mathbf{o}^\star \rangle\delta + \frac{4\varepsilon M\delta}{\sqrt{k}} + n^2\delta^2 M \qquad \text{(by element-wise non-positive Hessian)}$$
$$\leq \langle \nabla F(\mathbf{x}(t)), \mathbf{v}(t) \rangle\delta + 2\varepsilon M\delta + \frac{4\varepsilon M\delta}{\sqrt{k}} + n^2\delta^2 M \qquad \text{(by Lemma E.2)}$$
$$\leq 2\langle \nabla F(\mathbf{x}(t)), \widehat{\mathbf{v}}(t) \rangle\delta + 2\varepsilon M\delta + \frac{5\varepsilon M\delta}{\sqrt{k}} + n^2\delta^2 M \qquad \text{(by Eq. (16))}$$
$$\leq 2\left(F(\mathbf{x}(t+\delta)) - F(\mathbf{x}(t))\right) + 2\varepsilon M\delta + \frac{5\varepsilon\delta}{\sqrt{k}} + 2n^2\delta^2 M. \qquad \text{(by approximate linearity)}$$

with probability at least $1 - \frac{\varepsilon\eta}{n^2+1}$. To sum the above inequalities over $t = 0, \delta, \ldots, 1$ and apply the union bound on the probability , we conclude that

$$F(\mathbf{o}(0)) - F(\mathbf{o}(1)) \leq 2\left(F(\mathbf{x}(1)) - F(\mathbf{x}(0))\right) + 5\varepsilon M,$$

with probability at least $1 - \eta$. $\qquad\qquad\square$

Finally, the query complexity of Algorithm 3 is identical to that of Algorithm 1. We complete the proof of Theorem F.1 by combining Lemma F.2 and the following ronding lemma which is similar to Lemma 3.2.

**Lemma F.3 (Rounding scheme for non-monotone case).** *Let $\epsilon, \eta \in (0, 1)$, and $\mathcal{P}$ be a constant. Suppose for any monotone $k$-submodular function $f'$, Algorithm NFW$(f', \mathcal{P}, \epsilon, \eta)$ outputs a solution $\mathbf{x} \in \Delta_k^n$ that satisfies $F(\mathbf{x}) \geq \alpha \max_{\mathbf{y} \in \Delta_k^n, \mathbf{y} \sim \mathcal{P}} F(\mathbf{y})$ Then for any $\varepsilon > 0$, there exists a rounding scheme that outputs a solution $\mathbf{s} \in \{0, \ldots, k\}^n$ with $\mathbf{s} \sim \mathcal{P}$, that is*

- *$\alpha$-approximate when $\mathcal{P}$ is single matroid constraint, i.e., $f(\mathbf{s}) \geq \alpha \cdot \max_{\mathbf{s'} \in \{0,\ldots,k\}^n, \mathbf{s'} \sim \mathcal{P}} f(\mathbf{s'})$, with calling NFW one time and calling $\mathcal{O}_{\mathcal{P}}$ at most $O(Nn^2)$ times;*

- *$\alpha(1 - \varepsilon)$-approximate when $\mathcal{P}$ is $l = O(1)$ knapsack constraints, with calling NFW$, O_f, \mathcal{O}_{\mathcal{P}}$ at most $O\left(k^{\mathrm{poly}(1/\varepsilon)} n^{\mathrm{poly}(1/\varepsilon)}\right)$ times;*

- *$\left(\frac{0.6\alpha}{b}(1 - \varepsilon)\right)$-approximate when $\mathcal{P}$ is the intersection of $b$ matroid constraints and $l = O(1)$ knapsack constraints, with calling NFW$, O_f, \mathcal{O}_{\mathcal{P}}$ at most $O\left(k^{\mathrm{poly}(1/\varepsilon)} n^{\mathrm{poly}(1/\varepsilon)}\right)$ times.*

The proof is omitted as the proof of Lemma 3.2 does not require the monotonicity of the objective function. The only difference is that we use NFW for the non-monotone case.

# G  HARDNESS FOR THE INTERSECTION OF $O(1)$ KNAPSACKS AND $b$ MATROIDS

**Theorem G.1 (Hardness for the intersection of $b$ matroids).** *There exist instances of $k$-submodular maximization with support constraints, $\max\{f(\mathbf{s}), \mathbf{s} \in \mathcal{P}\}$, where $\mathcal{P}$ is intersection of $b$ matroids, any algorithm with better than $O(\log b/b + \varepsilon)$ approximation ratio for this problem would require exponentially many value queries for any $\varepsilon > 0$, unless $P = NP$.*

*Proof.* Consider the monotone $k$-submodular function $g : \{0, 1, 2\}^n \to \mathbb{R}_{\geq 0}$, defined as

$$g(\mathbf{s}) = f(S_1) + \varepsilon f(S_2),$$

where $f : 2^n \to \mathbb{R}_{\geq 0}$ is a monotone submodular function, $\varepsilon \in \mathbb{R}^+$, $S_1 = \{i : \mathbf{s}_i = 1\} \in 2^n$ and $S_2 = \{i : \mathbf{s}_i = 2\} \in 2^n$. We set $\varepsilon$ to be sufficiently small such that

$$\min_i f(N) - f(N \setminus \{i\}) \geq \varepsilon \left(\max_i f(i) - f(\varepsilon)\right).$$

This ensures that the optimal solution $\mathbf{o} \in \{0, 1, 2\}^n$ satisfies that $\mathbf{o}_i \in \{0, 1\}$ for all $i \in [n]$. Otherwise, we can improve the function value by changing the value $\mathbf{o}_i$ from 2 to 1 without conflict with the support constraint. In other words, maximizing the $k$-submodular function $g$ subject to any support constraint is equivalent to maximizing the submodular function $f$ with the same constraint.

It is well-known that unless $P = NP$, there is no approximation algorithm better than $O(\log b/b)$ for $b$-dimensional matching (see (Hazan et al., 2006)). Hence, there is no better approximation algorithm for submodular maximization subject to the intersection of $b$ matroids constraint (Lee et al., 2010b). Therefore, the hardness result of $O(\log b/b)$ also holds for constrained $k$-submodular maximization. $\square$

