# OpenReview forum: "Improved Approximation Algorithms for $k$-Submodular Maximization via Multilinear Extension"
_ICLR.cc/2025/Conference — ICLR 2025 Spotlight_

### Official Review · Reviewer_HNQi · 2024-10-31

**Soundness:** 3
**Presentation:** 3
**Contribution:** 2
**Rating:** 6
**Confidence:** 3

**Summary:**

This paper addresses the problem of maximizing $k$-submodular functions under various constraints. By generalizing the multi-linear extension commonly used for standard submodular functions to apply to $k$-submodular functions, the authors introduce unified Frank-Wolfe-type frameworks to tackle these problems in a continuous domain. The proposed algorithm attains an approximation ratio of $1/2$ for monotone $k$-submodular optimization and $1/3$ for nonmonotone $k$-submodular optimization.

**Strengths:**

1. The problem of k-submodular maximization is important in many fields of machine learning and aligns well with the conference’s broader focus.
2. The paper provides solid and clear proof analysis and proposes a unified algorithm that achieves better approximation ratios compared with the previous method.
3. The proof sketch of Lemma 3.3 is presented clearly and highlights the technical novelty of the proof.

**Weaknesses:**

1. As has been highlighted in [1], the $k$-submodular functions can be reduced to general submodular functions under partition matroid.  Given this reduction, the idea of defining the multi-linear extension for $k$-submodular functions is less novel.
2. Since this is a theoretical work with no experimental results, the primary contributions should ideally lie in offering new technical skills or insights to the submodular optimization community. However, most of the proof analysis appears standard.

-[1]. Satoru Iwata, Shin-ichi Tanigawa, and Yuichi Yoshida. Improved approximation algorithms for
k-submodular function maximization.

**Questions:**

The paper is clearly written. My main question concerns the proof of Lemma 3.3. I understand that the best-known lower bound for the monotone case is $1/2$, but could you please give some intuition of why the algorithm can't achieve a better bound (e.g. $1-1/e$) by explaining some key aspects of the proof in Lemma 3.3? Additionally, I noticed that the query complexity of the algorithm, even for the monotone case, is somewhat inefficient. Would using a different concentration inequality, such as Hoeffding's inequality instead of the Chernoff bound, potentially improve the query complexity?

---

> ### Author Response · Authors · 2024-11-21
>
> We thank the reviewer for their thoughtful feedback. We are encouraged by the recognization of the importance of the $k$-submodular maximization problem, the clarity of our proofs, and the unified algorithm achieving better approximation ratios. Below, we address the questions raised.
>
> >...the k-submodular functions can be reduced to general submodular functions under partition matroid. Given this reduction, the idea of defining the multi-linear extension for $k$-submodular functions is less novel
>
> We believe it is non-trivial to reduce non-negative $k$-submodular functions to general non-negative submodular functions under a partition matroid. The statement in [Iwata et al., 2016]—"The $k$-submodular function maximization problem is closely related to the submodular function maximization with a partition matroid constraint"—does not imply such a reduction exists.
>
> One possible attempt of reduction is as follows. We define the domain as $\\bar{\Delta}\_k\^n\subseteq \\{0,1\\}\^{nk}$ where $\bar{\Delta}\_k = \\{x \in \{0,1\}\^{k} : \sum\_{j=1}\^k x_j \leq 1\\}$, and define a function $\bar{f}: \bar{\Delta}\_k\^n\to \mathbb{R}$ as $\bar{f}(S) = f(\mathbf{s})$, where $\mathbf{s}$ is defined as $\mathbf{s}\_i = j$ if there exists a unique element $e\_{i,j} \in S$ and $\mathbf{s}_i = 0$ otherwise. However, we may not be able to extend the domain of such a submodular function to $\\{0,1\\}^{nk}$ without violating non-negativity or monotonicity, even for the simplest case of $k = 2$. Specifically:
> 1. As shown in Lemma 2 of [Singh et al., 2012], there exists a non-negative 2-submodular function, for which no extension is both non-negative and submodular.
> 2. Furthermore, Lemma 3 of [Singh et al., 2012] demonstrates that there exists a monotone, non-negative 2-submodular function, for which no extension is non-negative, monotone, and submodular.
>
> These results highlight that $k$-submodular functions cannot generally be reduced to submodular functions under partition matroids, and the development of a multilinear extension tailored specifically for $k$-submodular functions remains new.
>
> We have added this explaination in the revised version; see Appendix A.3 in our [updated manuscript](https://openreview.net/pdf?id=EPHsIa0Ytg).
>
>
> [Iwata et al., 2016] Improved Approximation Algorithms for k-Submodular Function Maximization, Satoru Iwata, Shin-ichi Tanigawa, Yuichi Yoshida.
>
> [Singh et al., 2012] On Bisubmodular Maximization. Ajit P. Singh, Andrew Guillory, Jeff Bilmes.
>
>
>
> >...give some intuition of why the algorithm can't achieve a better bound (e.g. $1−1/e$) by explaining some key aspects of the proof in Lemma 3.3?
>
> The main reason why the approximation ratio is not as good as $1−1/e$ for $k$-submodular maximization is due to the change of domain from cube $[0,1]^{nk}$ to a corner $\Delta_k^n$. To illustarte this, we first recap why FW can find $1-1/e$ approximate solution for submodular optimization. To prove the approximate ratio, we link the current solution $x^k$ and optimal solution $x^\star$ by auxillary point $x^\star \vee x^k$ by coordinate-wise maximum operation, then showing one step improvement as  $F(x^{k+1}) - F(x^\star)\gtrsim (1-\delta)(F(x^k)- F(x^\star))$ with step-size $\delta$ and solution $x^k$ at $k$-step of FW method. By accumulation, we have $F(x^{K}) \gtrsim (1-1/e) F(x^\star)$. However, the function value of the auxillary point $x^\star \vee x^k$ by coordinate-wise maximum operation may not have definition for $k$-submodular. Instead, in the proof of Lemma 3.3, we use new auxilary point $o(t) = x(t) + (1 − t)o^\star$ that lies in domain $\Delta_k^n$, and show one step improvement $F(x(t + \delta)) − F(x(t)) \gtrsim F(o(t)) − F(o(t + \delta))$, which results in $F(x^{K}) \gtrsim 1/2 F(x^\star)$.
>
>
> > Would using a different concentration inequality, such as Hoeffding's inequality instead of the Chernoff bound, potentially improve the query complexity?
>
> We agree that using alternative concentration inequalities, such as Hoeffding's inequality (instead of the Chernoff bound), might improve the query complexity during the first phase of maximizing the continuous function. However, the overall query complexity of the algorithm could be dominated by the rounding procedure, which remains polynomial even if we have an improved concentration. For instance, finding a continuous solution under a knapsack constraint requires $O(\frac{kn^6\log\frac{n\varepsilon}{\eta}}{\varepsilon^3})$ queries, while the rounding process to obtain a discrete solution necessitates an additional $O(k^{poly(1/\varepsilon)} n^{poly(1/\varepsilon)})$ queries.  Additionally, our primary focus in this work is on achieving the best approximation ratio rather than minimizing query complexity. We believe that exploring such refinements to improve query complexity in the first phase will be an interesting direction if the first phase query becomes dominant in some future settings.

---

> > ### Comment · Reviewer_HNQi · 2024-12-01
> > **Reviewer Comment**
> >
> > Thanks for the clarification! I don't have any further concerns.

---

### Official Review · Reviewer_t6jy · 2024-11-05

**Soundness:** 3
**Presentation:** 4
**Contribution:** 3
**Rating:** 8
**Confidence:** 4

**Summary:**

This paper considers the problem of maximization of $k$-submodular functions, where elements in the ground set can be selected into one of $k$ sets. $k$-submodular functions are a generalization of ordinary submodular set functions, which correspond to the special case $k=1$. For the maximization of ordinary submodular set functions, the continuous multilinear extension has been used in order to yield algorithms with better approximation guarantees compared to combinatorial approaches such as greedy algorithms. In contrast, only combinatorial approaches have previously been proposed for $k$-submodular function maximization. Inspired by the effectiveness of continuous approaches in the ordinary submodular function case, this paper extends the definition of multilinear extension to $k$-submodular functions, and proposes algorithms using this multilinear extension to achieve better approximation guarantees that those existing for a variety of constraints (see Table 1). Their algorithms use Frank-Wolfe types of methods, which is different than continuous algorithms used for ordinary submodular functions.

**Strengths:**

- $k$-submodular functions have been a topic of recent interest, and this paper makes an important contribution by introducing an extended multilinear extension and proposing algorithms which have better approximation guarantees using the multilinear extension.
- They developed algorithms with theoretical performance guarantees stronger than those existing in the literature. Further, at least some of their guarantees matched the best known lower bound of Iwata et al. (2016).
- They used Frank-Wolfe type of methods instead of simply extending the most standard algorithms using the multilinear extension for standard submodular functions such as that of Calinescu et al. [2011] (see reference below). They further explain the technical challenges that arise when simply trying to extend those standard approaches. They also had to use an alternative method of rounding. So their theoretical results do not seem trivial to me.
- Paper is very well-written and clear.

Calinescu, Gruia, et al. "Maximizing a monotone submodular function subject to a matroid constraint." SIAM Journal on Computing 40.6 (2011): 1740-1766.

**Weaknesses:**

- There is no experimental evaluation of the algorithms, and in fact the algorithms might not be very practical to implement. Multilinear extension algorithms are often much slower and less practical compared to combinatorial ones for ordinary submodular functions, so one would probably expect that to also be the case in this setting.
- This paper extends and combines ideas from many papers in the literature. For example, the ordinary submodular function multilinear extension is extended to the $k$-submodular case, and in addition Frank-Wolfe style algorithms have been used for ordinary submodular functions. One con then might be that there is not a huge amount of novelty, but I think it is still plenty sufficient for publication.

**Questions:**

* Is the definition of $k$-submodular presented in the paper the typical definition? And if it is not, is it clearly equivalent? I recall seeing different definitions of $k$-submodular in related work.

---

> ### Author Response · Authors · 2024-11-21
>
> Thank you for reviewing our manuscript. We are encouraged by your recognition of the importance, novelty, and clarity of our work.
>
> > Is the definition of $k$-submodular presented in the paper the typical definition? And if it is not, is it clearly equivalent?
>
> We appreciate the question regarding the definition of $k$-submodularity. Yes, both the definition in our paper (see Eq. (1)) and the one from [Sakaue (2017)] (as noted in the footnote on Page 1) are typical definitions for $k$-submodular and they are equivalent. To clarify, let us outline the equivalence with the definition on Page 2 of [Sakaue (2017)]:
>
>
> **Definition 1 (from  [Sakaue (2017)])**
> - The domain is represented as $(k+1)^V := \{(X_1, \ldots, X_k) \mid X_i \subseteq V, X_i \cap X_j = \emptyset \text{ for } i \neq j\}$, where $V$ is a ground set.
> - A function $f: (k+1)^V \to \mathbb{R}$ is called $k$-submodular if for any $x = (X_1, \ldots, X_k)$ and $y = (Y_1, \ldots, Y_k)$, it satisfies:
> $f(x) + f(y) \geq f(x \cap y) + f(x \cup y),$
>   where:
>   - $(x \cap y)_i = X_i \cap Y_i$,
>   - $(x \cup y)\_i = X_i \cup Y_i \setminus \bigcup_{j \neq i} (X_j \cup Y_j)$.
>
>
> **Definition 2 (from our paper)**
> - The domain is represented as $\{0, 1, \ldots, k\}^n$, where $0$ represents the null element and each element of the vector specifies a label for one of $k$ disjoint sets.
> - A function $f: \{0, 1, \ldots, k\}^n \to \mathbb{R}_{\geq 0}$ is called $k$-submodular if for any $s, t \in \{0, 1, \ldots, k\}^n$, it satisfies:
> $f(s) + f(t) \geq f(\min_0(s, t)) + f(\max_0(s, t)),$
>   where:
>   - $(\min_0(s, t))_i =
>       \begin{cases}
>       0 & \text{if } s_i t_i \neq 0 \text{ and } s_i \neq t_i, \\\\
>       \min(s_i, t_i) & \text{otherwise},
>       \end{cases}$
>   - $(\max_0(s, t))_i =
>       \begin{cases}
>       0 & \text{if } s_i t_i \neq 0 \text{ and } s_i \neq t_i, \\\\
>       \max(s_i, t_i) & \text{otherwise}.
>       \end{cases}$
>
> #### Mapping Between Domains:
> - In Definition 1, the domain is $(k+1)^V$, which represents $k$ disjoint subsets $X_1, \ldots, X_k$ of a ground set $V$.
> - In Definition 2, the domain is $\{0, 1, \ldots, k\}^n$, where each element $i \in [n]$ belongs to one of $k$ disjoint sets, represented by its label $1, \ldots, k$, or is null ($0$).
>
> The two domains are equivalent since labeling each element in $V$ corresponds directly to assigning it to a subset $X_i$ (or $0$ if it belongs to none).
>
> #### Operations ($\cup, \cap$ vs. $\min_0, \max_0$):
> - **Intersection ($\cap$ in Definition 1 and $\min_0$ in Definition 2):**
>   - For Definition 1, $(x \cap y)_i = X_i \cap Y_i$.
>   - For Definition 2, $(\min_0(s, t))_i = \min(s_i, t_i)$ when $s_i, t_i \in \{0, i\}$, which aligns with the intersection of sets for corresponding labels. If $s_i$ and $t_i$ are different and nonzero, the result is $0$, corresponding to the empty intersection.
>
> - **Union ($\cup$ in Definition 1 and $\max_0$ in Definition 2):**
>   - For Definition 1, $(x \cup y)\_i = X_i \cup Y_i \setminus \cup_{j \neq i}(X_j \cup Y_j)$, which ensures disjointness.
>   - For Definition 2, $(\max_0(s, t))_i = \max(s_i, t_i)$ when $s_i, t_i \in \{0, i\}$, which aligns with the union of sets. If $s_i$ and $t_i$ are different and nonzero, the result is $0$, ensuring disjointness.
>
> #### Submodular Inequality:
> - Both definitions use the inequality:
> $f(x) + f(y) \geq f(x \cup y) + f(x \cap y),$
>   which holds in both formulations because the operations $\cup, \cap$ in Definition 1 are equivalent to $\max_0, \min_0$ in Definition 2, and the domains and function mappings are equivalent.
>
>
> We have revised our manuscript (see [revised manuscript](https://openreview.net/pdf?id=EPHsIa0Ytg)), where we highlight the existence of this equivalent definition of $k$-submodular in the introduction section. We also add this definition and explain its equivalence to our definition (Eq. (1)) in the updated Appendix A.2.

---

### Official Review · Reviewer_pvDA · 2024-11-08

**Soundness:** 3
**Presentation:** 4
**Contribution:** 2
**Rating:** 8
**Confidence:** 3

**Summary:**

This paper addresses the k-submodular maximization problem under matroid and knapsack constraints. k-submodular maximization is a generalization of submodular maximization where each element of an n-dimensional vector can take values from {0, ..., k}. The objective is to maximize a submodular function defined on these vectors, subject to given constraints.

The authors apply the multilinear extension technique—previously used in submodular maximization but novel to k-submodular maximization—achieving an improved approximation factor for this problem, particularly under the d-knapsack constraint. They have improvements for other constraints too, but most improvements are minor except for this specific constraint.

**Strengths:**

The main strength lies in applying the multilinear extension to k-submodular maximization, which could inspire similar approaches in related problems. Additionally, the improvement in the approximation factor under the d-knapsack constraint is noteworthy.

**Weaknesses:**

Although the algorithm provides a good approximation factor, its practical applicability is limited due to potentially high running times, and no experimental results are provided to assess real-world performance. Additionally, while the first paragraph of introduction mentions applications, it’s unclear how relevant this problem is for the ML community, as it’s not as widely studied as submodular maximization, which may hold more appeal for ML research.

## Minor comments:
- On page 5, you mention that Niu et al. achieved a 1/3-approximation ratio for the non-monotone case under a single matroid constraint, but a worse result appears on page 2, Table 1.
- On page 2, the notation for min_0 and max_0 is inconsistent—sometimes with 0 as a subscript, other times appearing below them. Please standardize.

**Questions:**

Could you compare your results with those for submodular maximization to clarify the gap between this work and existing results for multilinear extension in submodular maximization?

---

> ### Author Response · Authors · 2024-11-21
>
> Thank you for your positive and encouraging feedback. We appreciate the recognition of our work’s contribution and will address the questions raised in your comments.
>
> > It’s unclear how relevant this problem is for the ML community
>
> While submodular maximization may be more famous in the ML community, many applications of it could be extended to $k$-submodular maximization. One example is diversity, where the selection needs to balance multiple sources.
> * Feature selection: In machine learning, feature selection is the process of identifying a subset of features that are most relevant to a given task. $k$-submodular maximization can be used to find a diverse set of features that maximizes the performance of a model.
> * Active learning: Active learning is a technique for selecting the most informative data points to label, which can help to reduce the cost of labeling data. $k$-submodular maximization can be used to select a diverse set of data points that are likely to provide the most information about the underlying model.
> * Recommendation systems: Recommendation systems are used to provide personalized recommendations to users. $k$-submodular maximization can be used to select a diverse set of items that are likely to be of interest to a given user.
>
> There are many other applications. When we mention sensor placement in the manuscript, this is related to data acquisition in machine learning. Determining the optimal placement of sensors to collect the most informative data for training a model. It is also useful in anomaly detection, where one trategically places monitoring agents within a network to maximize the chances of detecting anomalies. Meanwhile, $k$-submodular optimization is useful for resource allocation tasks, which is relevant in several ML senarios. In distributed computing, one may assign tasks to a limited number of computing nodes to optimize ML training performance and energy consumption. In cloud ML, one may allocate different types of virtual machines or containers to meet varying workloads while minimizing costs.
>
> We have added this discussion in the revised version; see Appendix A.1 in [revised manuscript](https://openreview.net/pdf?id=EPHsIa0Ytg).
>
> > Could you compare your results with those for submodular maximization to clarify the gap between this work and existing results for multilinear extension in submodular maximization?
>
> Below we clarify the distinction between our work and existing results on the multilinear extension in submodular maximization, which we have addressed in lines 253–260.
>
> Specifically, for submodular maximization, the results for **monotone cases** using multilinear extension-based algorithms are as follows:
> 1. A $(1-1/e-\varepsilon)$-approximation for $O(1)$ knapsacks;
> 2. A $(1-1/e-\varepsilon)$-approximation for a single matroid;
> 3. A $(0.6(1-1/e)/b - \varepsilon)$-approximation for the intersection of $O(1)$ knapsacks and $b$ matroids, all achievable in polynomial time with respect to $n$.
>
> For **non-monotone cases**, these algorithms achieve:
> 1. A $(0.401-\varepsilon)$-approximation for $O(1)$ knapsacks;
> 2. A $(0.401-\varepsilon)$-approximation for a single matroid;
> 3. A $(0.24/b - \varepsilon)$-approximation for the intersection of $O(1)$ knapsacks and $b$ matroids, also in polynomial time with respect to $n$.
>
> In comparison, for **$k$-submodular maximization**, multilinear extension-based algorithms yield the following results for **monotone cases**:
> 1. A $(1/2-\varepsilon)$-approximation for $O(1)$ knapsacks;
> 2. A $(1/2-\varepsilon)$-approximation for a single matroid;
> 3. A $(0.3/b - \varepsilon)$-approximation for the intersection of $O(1)$ knapsacks and $b$ matroids, all achievable in polynomial time with respect to $n$ and $1/\varepsilon$.
>
> For **non-monotone cases**, the results are:
> 1. A $(1/3-\varepsilon)$-approximation for $O(1)$ knapsacks;
> 2. A $(1/3-\varepsilon)$-approximation for a single matroid;
> 3. A $(0.2/b - \varepsilon)$-approximation for the intersection of $O(1)$ knapsacks and $b$ matroids, also in polynomial time with respect to $n$.

---

> > ### Comment · Reviewer_pvDA · 2024-11-25
> >
> > Thank you for the explanation. Given that, I've increased my score.

---

### Official Review · Reviewer_Tvbp · 2024-11-11

**Soundness:** 3
**Presentation:** 3
**Contribution:** 3
**Rating:** 8
**Confidence:** 3

**Summary:**

This paper studies monotone and non-monotone k-submodular maximization subject to various constraint systems. Prior work on this topic has studied combinatorial algorithms only -- this paper introduces a multilinear extension for k-submodular functions. Despite the power of the continuous approach for 1-submodular functions, it hasn't been applied to the k-submodular case. Via this method, the authors obtain improved approximation ratios in several constraint regimes, most notably O(1) knapsacks, where they achieve an asymptotically tight ratio.

**Strengths:**

- In some sense, developing continuous algorithms for the k-submodular case seems like a natural generalization of the 1-submodular methods. However, there were a number of challenges the authors had to adress, which is likely why prior work had avoided this direction. Everything looks easy in hindsight.
- Specifically, the way the authors addressed a certain feasibility issue (discussed on page 4) by shifting the optimization target from o^* is relatively novel. It is a little surprising to me, not that this method obtains a constant factor, but that it gets the optimal ratio in some settings (O(1) knapsacks). Other generalizations (such as rounding, approximate linearity, etc.) seem more straightforward.
- For non-monotone functions, the ratios aren't tight, but significant improvements in state-of-the-art are obtained.

**Weaknesses:**

- Notation departs substantially from most of the other k-submodular papers I've seen. It took a bit of thought to see that everything is equivalent. Likely this could have been introduced in a more intuitive way that would make the paper more accessible.
- The main text is really an extended abstract, with no proofs. And instead, arguments why the methods are interesting. In general, I would like to see at least some part of the technical arguments condensed in the main text -- but this is more a critique of the publication / reviewing model. Often, mistakes are found later (obviously didn't have time to check the 30-page appendix in full detail).
- Algorithms are of theoretical interest only, of order n^{poly(1/\epsi}. It is difficult to imagine a scenario where one would want to try to implement these algorithms. Thus, their ability to tackle big data k-submodular instances of problems relevant to the ML community is limited to non-existent.

**Questions:**

It seems like non-monotone k-submodular optimization just uses monotone methods with a partial monotonicity property implied by k-submodular. But these results don't appear to be tight, and the problem seems to be much less well understood. Can the authors shed any light on this?

---

> ### Author Response · Authors · 2024-11-21
>
> Thank you for your positive feedback. We are glad that you appreciate the technical novelty of our work.
>
>
> > Notation departs substantially from most of the other $k$-submodular papers ...could have been introduced in a more intuitive way...
>
> Thank you for the suggestion. It is indeed important to make the notations more friendly to the audience who are already familiar with the other definition of $k$-submodular functions (as noted in the footnote on Page 1). We have revised our manuscript (see [revised version](https://openreview.net/pdf?id=EPHsIa0Ytg)), where we highlight the existence of an equivalent definition of $k$-submodular in the introduction section. We also add this definition and explain its equivalence to our definition (Eq. (1)) in the updated Appendix A.2.
>
>
> > It seems like non-monotone $k$-submodular optimization just uses monotone methods with a partial monotonicity property implied by $k$-submodular. But these results don't appear to be tight, and the problem seems to be much less well understood. Can the authors shed any light on this?
>
> It is indeed important to understand the gap between our result and negative results. We do have some intuitions for why the results for non-monotone $k$-submodular optimization are not tight.
>
> - First, we note that even for $k=1$, there remains a longstanding open problem in closing the gap between the best known 0.401-approximation algorithm achieved by a variant of continuous greedy [Buchbinder et al., 2024] and the 0.478 inapproximability result for the non-monotone submodular optimization problem. Since $k$-submodular optimization generalizes submodular optimization, we anticipate that achieving tight results for the non-monotone $k$-submodular case will be at least equally (or even more) challenging.
>
>
> - Second, most of the recent advances in continuous methods for submodular optimization concentrate on continnuous greedy and FW-type methods in the monotone case. In fact, FW acts as a continuous analogue of the greedy algorithm, mimicking its selection of high marginal gain directions and inheriting its tight $(1−1/e)$-approximation effectiveness. As a comparision, the marginal gain is hard to estimate for non-monotone case, and the linear surrogate often fails to capture the non-linear interactions between elements. This indicates that it is less likely to achieve tightness through FW-type algorithm.
>
>
> [Buchbinder et al., 2024] Constrained Submodular Maximization via New Bounds for DR-Submodular Functions, Niv Buchbinder and Moran Feldman, STOC 2024.

---

> > ### Comment · Reviewer_Tvbp · 2024-11-27
> >
> > Thank you for the response. I will maintain my score.

---

### Meta-Review · Area_Chair_6hXW · 2024-12-21

**Metareview:**

The paper studies the problem of maximizing a k-submodular objective function subject to constraints such as cardinality, matroid, and knapsack constraints. The problem is a generalization of submodular maximization, which is the special case k=1, with additional applications. This work is the first to extend the well-known approach based on the multilinear extension and continuous optimization from submodular functions to k-submodular functions. The resulting algorithms achieve improved approximation guarantees in several settings.

The reviewers appreciated the main contributions and generally found the theoretical contribution to be strong and novel. The improvement in the approximation for certain constraints such as multiple knapsack constrains is significant. A notable weakness raised by the reviewers is that the contribution is primarily of theoretical interest, since the running time is prohibitive in practice. Overall, the paper makes a valuable theoretical contribution to the area of submodular maximization.

**Additional Comments On Reviewer Discussion:**

In addition to the main concern that the algorithms are inefficient, the reviews also raised some concerns regarding the exposition and the practical motivation for studying k-submodular maximization problems. The authors revised the paper to address these comments.

---

### Decision · Program_Chairs · 2025-01-22

Accept (Spotlight)